# Southern Ocean CO$_2$ outgassing and nutrient load reduced by a well-ventilated glacial North Pacific

Madison G. Shankle [1] ✉, Graeme A. MacGilchrist [1], William R. Gray [2], Casimir de Lavergne [3], Laurie C. Menviel [4], Andrea Burke [1] & James W. B. Rae [1]

Southern Ocean biogeochemistry impacts global nutrient distributions, carbon cycling, and climate, motivating study of its underlying controls across different climate states. Today, poorly-ventilated North Pacific waters supply the majority of carbon and nutrients upwelling in the Southern Ocean, outpacing biological carbon uptake and fueling CO$_2$ outgassing. Reducing this supply is both central to glacial CO$_2$ theories involving reduced outgassing and well-supported by paleo-proxy reconstructions. While past studies emphasize physical processes (reduced upwelling, enhanced stratification), we propose a complementary mechanism where the carbon/nutrient load of waters feeding the Southern Ocean surface is reduced remotely, prior to being upwelled. Comparing glacial North Pacific and Southern Ocean proxy records, alongside Earth System Model simulations, we show that ventilating the glacial North Pacific reduces the carbon/nutrient content of waters supplying the Southern Ocean surface and Subantarctic CO$_2$ outgassing. This highlights an inter-hemispheric influence on Southern Ocean biogeochemical conditions that could modulate glacial-interglacial CO$_2$ variability.

The Southern Ocean is an important nexus for biogeochemical cycling, connecting the ocean basins as well as the deep and surface layers of the ocean. The biogeochemistry of Southern Ocean surface waters supports thriving local ecosystems[1], influences global ocean nutrient distribution[2,3], and regulates ocean-atmosphere carbon exchange[4–6]. Together, these processes can influence the global carbon cycle through their impact on carbon dioxide (CO$_2$) outgassing in this region.

Today, and in interglacial periods in Earth's past, wind-driven upwelling supplies an over-abundance of major nutrients to Southern Ocean surface waters, where biological production – limited by availability of light[7] and iron[8] (a minor nutrient) – is insufficient to entirely extract the accompanying upwelled carbon and thus counteract the evasion of carbon dioxide (CO$_2$) from the deep ocean to the atmosphere. (Note that this refers not to the Southern Ocean's present-day strong uptake of anthropogenic carbon but rather to its pre-industrial unperturbed state, in which it was a net source of natural carbon[9–11].) This CO$_2$ evasion[12,13] – compounded by the loss of associated unused or "preformed" nutrients back into the abyss via Antarctic Bottom Water, losing their ability to drive biological carbon uptake elsewhere in the surface ocean – is why the Southern Ocean is described as being a "leak" of CO$_2$ to the atmosphere. Improving biology's ability to "keep up" with the incoming carbon/nutrient supply in the past could have stemmed this leak of CO$_2$ and has been invoked in many theories for atmospheric CO$_2$ drawdown during glacial periods[5,14–17]. Besides increasing export production rates, for instance via iron

[1]School of Earth and Environmental Sciences, University of St Andrews, St Andrews, United Kingdom. [2]Laboratoire des Sciences du Climat et de l'Environnement (LSCE/IPSL), CEA-CNRS-UVSQ, Université Paris-Saclay, Gif-sur-Yvette, France. [3]LOCEAN Laboratory, Sorbonne Université-CNRS-IRD-MNHN, Paris, France. [4]Climate Change Research Centre, The Australian Centre for Excellence in Antarctic Science, University of New South Wales, Sydney, NSW, Australia. ✉e-mail: mgs23@st-andrews.ac.uk

fertilization[18–21], $CO_2$ evasion to the atmosphere could have been mitigated by any mechanism which reduced the supply of carbon and nutrients to the Southern Ocean surface. Under such conditions and all else held equal, the same levels of productivity would fix a greater proportion of the reduced incoming carbon supply, reducing outgassing, and a reduced nutrient load of surface waters would ultimately lead to less preformed nutrients being lost to the abyss.

Thus far, proposed mechanisms of reducing the carbon and nutrient load of Southern Ocean surface waters during glacial times have primarily focused on reducing the physical transport of water to the surface (i.e., reduced upwelling and/or enhanced stratification, reducing the mixing of nutrient- and carbon-rich deep waters up from below). However, reducing the carbon and nutrient content of the upwelled water itself may also have played a critical role. It is therefore important to also consider potential changes in the composition of the water supplying the Southern Ocean surface.

Today and in interglacial states, Southern Ocean surface waters (Subantarctic and Antarctic Zones) receive the majority of their carbon and nutrients from the mid-depths (1–3 km) of the northern Pacific basin[12]. These mid-depths comprise Pacific Deep Water (PDW, ~1–3 km) that mixes with ambient water to form the Upper Circumpolar Deep Water (UCDW) that ultimately upwells into the Southern Ocean[22]. With overlying North Pacific Intermediate Water (NPIW) only ventilating the upper ~<1000 m, these mid-depth Pacific waters are poorly ventilated and largely isolated from exchange with the surface[23], allowing for the accumulation of significant amounts of remineralized carbon and nutrients. This is illustrated by modern profiles[24] of $CO_2$ outgassing potential (potential $pCO_2$, or "$PCO_2$", see Methods and Chen et al. (2022)[12]), nitrate, and phosphate spanning the Atlantic and Pacific basins (Fig. 1), which show nutrient- and carbon-

rich North Pacific mid-depth waters contrasting with relatively nutrient- and carbon-poor North Atlantic Deep Water (NADW). Figure 1 also illustrates the inclined isopycnals (tilted by near-freezing waters and wind-driven Ekman divergence at the surface) that link these mid-depths in both basins to the Southern Ocean surface (e.g., on neutral density surfaces - 27.5-28 kg m⁻³ in Fig. 1). The supply of these carbon rich, high-$PCO_2$ Pacific waters to the Southern Ocean – mainly through isopycnal diffusion[23,25] – in the present day has been described as the deep ocean's carbon exhaust[12]. Reducing the carbon and nutrient content of these mid-depth North Pacific waters feeding the Southern Ocean surface thus provides an alternative means of reducing the glacial Southern Ocean carbon and nutrient supply that could operate independently of, or alongside, reduced upwelling or mixing. Here, we explore this idea and the ability of North Pacific changes to induce important downstream effects on Southern Ocean biogeochemistry.

We first present biogeochemical proxy records from the North Pacific and Southern Ocean documenting a shared pattern of decreased surface nutrient levels during glacial periods[26,27]. We then show in a range of Earth System Models that reduced North Pacific surface nutrients is consistent with better ventilation (as evidenced by glacial proxy data[28,29]) and associated depletion of sub-surface nutrients. Finally, we show that these low-carbon, low-nutrient sub-surface North Pacific waters make their way south, where they directly translate into reduced nutrient load and outgassing rates in the Southern Ocean surface. These findings demonstrate an important connection between North Pacific and Southern Ocean biogeochemistry and suggest an interhemispheric mechanism for reducing the glacial carbon and nutrient load of Southern Ocean surface waters by far remote processes.

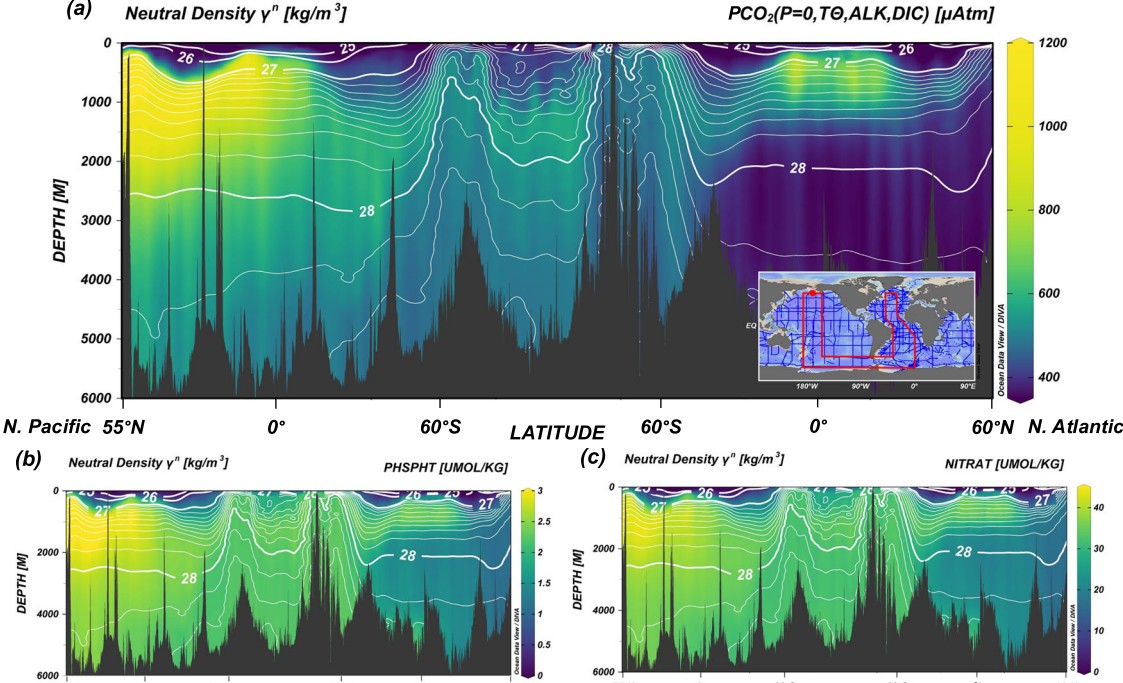

**Fig. 1 | Modern/interglacial Pacific and Atlantic basins show differing carbon and nutrient content.** Modern transects of carbon and nutrients from the Global Data Analysis Project (GLODAPv1)[24] give a representative picture of the pre-industrial/interglacial state of each basin, with a well-ventilated carbon- and nutrient-poor Atlantic (right end of each panel) contrasting with a poorly-ventilated carbon- and nutrient-rich Pacific (left end of each panel). **a** Potential $pCO_2$ ("$PCO_2$", where $pCO_2$ is the partial pressure of carbon dioxide, see Methods) (μatm), which quantifies a water parcel's tendency to outgas $CO_2$ when brought to the surface,

calculated from modern-day in-situ alkalinity, dissolved inorganic carbon (DIC), and potential temperature, referenced to 0 dbar. Anthropogenic carbon has been removed from DIC to represent pre-industrial-like carbon content. Modern (**b**) phosphate ($PO_4^{3-}$) and (**c**) nitrate ($NO_3^-$) concentrations (μmol kg-1) are not expected to differ significantly from pre-industrial levels. White contours on all panels show neutral density ($\gamma^n$) surfaces [kg m⁻³]. Figure made using Ocean Data View[58].

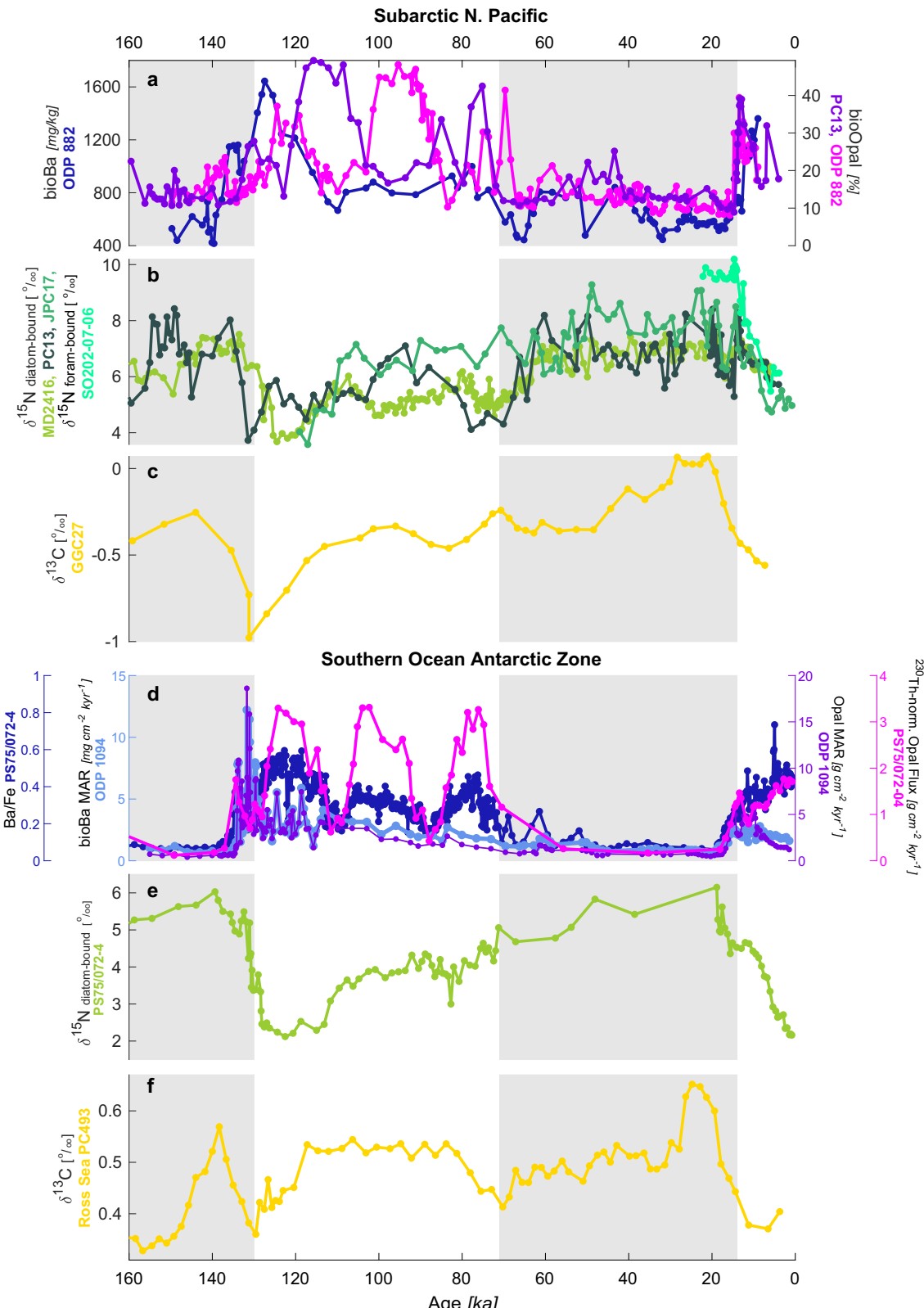

## Results

### Shared biogeochemistry in North Pacific and Southern Ocean

A reduced glacial supply of nutrients and carbon to Southern Ocean surface waters is recorded in proxy data over the last glacial cycle (Fig. 2). Reduced glacial biological productivity across the Antarctic Zone of the Southern Ocean is evidenced by proxies reflecting decreased export production (Fig. 2d). Such proxies include reduced fluxes and concentrations of biogenic barium and opal in marine sediments; various elemental ratios in sediments (e.g., reduced Ba/Fe and Ca/Fe reflecting reduced fluxes of biogenic barite and calcite to the seabed); reduced $^{231}$Pa/$^{230}$Th (reflecting reduced particle flux to depth); and reduced calcite fluxes (reflecting reduced export production of calcite-producing organisms)[19,30–38].

**Fig. 2 | Productivity and nutrient proxy records from the North Pacific and Southern Ocean share similar patterns over the Last Glacial Period.** Various proxies showing similar patterns in (**a**, **d**) primary productivity (sedimentary barium and opal concentrations, fluxes, and mass accumulation rates (MAR); purple and pink lines), (**b**, **e**) nutrient utilization ($\delta^{15}N$; green lines), and (**c**, **f**) ventilation (benthic foraminiferal $\delta^{13}C$; yellow lines) between the (**a**–**c**) North Pacific and (**d**–**f**) Southern Ocean. All y-axes are oriented to show increasing productivity, increasing (more complete) nutrient utilization, and better ventilation going upwards. Gray shading represents glacial conditions by demarcating marine isotope stages (MIS) 6 and 2-4. Data compiled from the following references (see Supplementary Fig. 1 for a map of the sites): ODP 882 (3255 m)[43,44]; PC13 (2393 m)[45]; MD2416 (2317 m)[46]; JPC17 (2209 m)[47]; SO202-07-06 (2345 m)[48] (note, "foram-bound $\delta^{15}N$" refers to *Neogloboquadrina pachyderma*); GGC27 (995 m)[51] (*Uvigerina* spp.); PS75/072-4 (3099 m)[41]; ODP 1094 (2850 m)[35]; PC493 (2077 m)[52] (*Cibicidoides wuellerstorfi*). The $\delta^{13}C$ records depict a 3- and 5-point moving average of the cited data in the North Pacific and Southern Ocean records, respectively, and are corrected for whole-ocean $\delta^{13}C$ change using the sea level record of Lea et al. (2002)[89] and a deglacial (Last Glacial Maximum to Holocene) whole-ocean $\delta^{13}C$ change of 0.34‰[90].

Alongside reduced productivity, the glacial Southern Ocean saw a more complete degree of nitrate utilization, with a greater proportion of the available nitrate pool being consumed by production. This is documented by elevated $\delta^{15}N$ values in diatom-, coral- and foraminifera-bound organic matter[39–42] (Fig. 2e). Biological productivity preferentially takes up nitrate with the lighter $^{14}N$ isotope, enriching the residual seawater nitrate pool in heavy $^{15}N$. As nutrient utilization becomes more complete – consuming a greater and greater proportion of the available nitrate pool – the residual heavy signal becomes increasingly incorporated into organic matter. Heavier $\delta^{15}N$ values under glacial conditions thus indicate enhanced nitrate utilization in Southern Ocean surface waters during glacial periods[39–42]. Critically, greater nitrate utilization (i.e., productivity consuming a greater proportion of the available nitrate pool) during glacial times can only be reconciled with reduced glacial productivity if the nitrate pool itself was smaller, signaling a reduced nutrient supply to the Southern Ocean surface in glacial times.

Remarkably similar glacial trends are also recorded in the far-flung North Pacific. Reduced productivity and export production are recorded throughout the subpolar gyre by reduced barium and opal fluxes[43–45] (Fig. 2a), as is enhanced nitrate consumption by heavier $\delta^{15}N$ values of foram- and diatom-bound organic matter[45–48] (Fig. 2b). This shared "Polar Twins" pattern[26] in Southern Ocean and North Pacific proxies has prompted considerable discussion of what process could simultaneously drive the same proxy patterns in two such far-flung regions. Previous explanations have focused on enhanced isolation of surface waters from nutrient-rich deep waters below, invoking the same physical processes (reduced upwelling or enhanced stratification) operating in both regions simultaneously during glacial times[26,27]. Little consideration, by contrast, has been given to linkages connecting these two regions and the potential importance of a common reduced subsurface nutrient and carbon content. In the North Pacific, reduced nutrient and carbon content in better-ventilated sub-surface waters has recently been demonstrated as a key means of reducing nutrient and carbon delivery to the North Pacific surface during the Last Glacial Maximum (LGM)[28]. Motivated by contemporary linkages between the North Pacific and Southern Ocean[12,13], we here explore the potential of North Pacific ventilation to simultaneously influence carbon and nutrient supply to the Southern Ocean surface.

**A reduced sub-surface North Pacific carbon/nutrient supply**

Proxy records show North Pacific surface waters exhibiting lower nutrients, higher salinity, and warmer temperatures in the LGM than today, which modeling results show to be consistent with better ventilation of and reduced nutrient content in sub-surface waters[28]. While reduced upwelling or mixing could also explain reduced surface nutrients, wind-driven upwelling (a dominant mode of nutrient delivery in the modern) was in fact likely increased in the North Pacific during glacial times, with Paleoclimate Modeling Intercomparison Project (PMIP) models consistently indicating substantially (~60%) greater wind stress curl over the North Pacific subpolar gyre under LGM conditions than in the pre-industrial[49,50]. Similarly, evidence of

enhanced regional intermediate water formation suggests increased convective mixing at this time[25,28], while increases in benthic carbon isotopes (Fig. 2c) reflect enhanced ventilation and influence by heavy-$\delta^{13}C$ surface waters[51,52]. Communication between the sub-surface and surface in the North Pacific, therefore, appears to have been increased under glacial conditions, thus requiring a reduced nutrient content of these sub-surface waters to explain reduced nutrient supply at the surface.

It is a common feature of Earth System Model simulations that such a reduction in sub-surface nutrient content can be achieved by an active meridional overturning circulation operating in the Pacific, as demonstrated in cGENIE[28] as well as higher-resolution models such as LOVECLIM and UVic[53,54]. Across these models, enhanced North Pacific Intermediate Water (NPIW) formation under glacial-like conditions consistently drives an equilibrium state in which well-ventilated sub-surface waters penetrate down to ~2000 m depth (hereafter referred to as "mid-depth" waters) (Supplementary Fig. 2 and Supplementary Table 1). Thus, where these mid-depths previously hosted poorly-ventilated carbon-rich waters isolated from the surface, the expanded NPIW cell has replaced these waters with relatively carbon- and nutrient-poor surface waters drawn from the low latitudes (Fig. 3 and Supplementary Fig. 3).

The same pattern of mid-depth carbon/nutrient reduction emerges in response to ventilation in these models, independent of varying configurations, boundary conditions, resolutions, and styles of forcing. Furthermore, mid-depth reductions in carbon and nutrients appear to be a robust response to varying degrees of enhanced ventilation of the North Pacific. The LOVECLIM-LGM simulation[54] presented in Fig. 3d, for example, simulates a weaker Pacific overturning than the other simulations (Supplementary Table 1), providing context of that study's estimated overturning at the LGM (~4 Sv). This simulation still sees strong reductions in mid-depth carbon and nutrients, and furthermore, may underestimate the true magnitude of glacial Pacific overturning, which remains poorly constrained. For example, Rae et al. (2020) found a Pacific overturning of ~8 Sv yielding the best model-data fit in c-GENIE simulations[28], and as such, the LOVECLIM-LGM output should be seen as broadly indicative rather than a definitive representation of LGM-like overturning. More importantly, the emergence of a consistent pattern across a range of strong and weak ventilation strengths (see also Chikamoto et al., 2012[55]) supports North Pacific ventilation as an effective means of reducing sub-surface carbon and nutrient content. Past studies have suggested other mechanisms by which such mid-depth Pacific anomalies can arise, such as changes in southern-sourced water mass characteristics impacting nutrient supply to the basin (Chikamoto et al., 2012). Such dynamics are not expected to be at play here, with the mid-depth anomalies clearly developing from the north due to the onset of northern ventilation (Supplementary Fig. 4).

Reducing the sub-surface carbon and nutrient pool also reduces local surface waters' carbon and nutrient content (Fig. 3), as it is these sub-surface waters which supply carbon and nutrients to the surface above[28]. The onset of greater convective mixing in the North Pacific may seem at odds with reduced surface nutrients and carbon, but it is important to distinguish between the transient and equilibrated

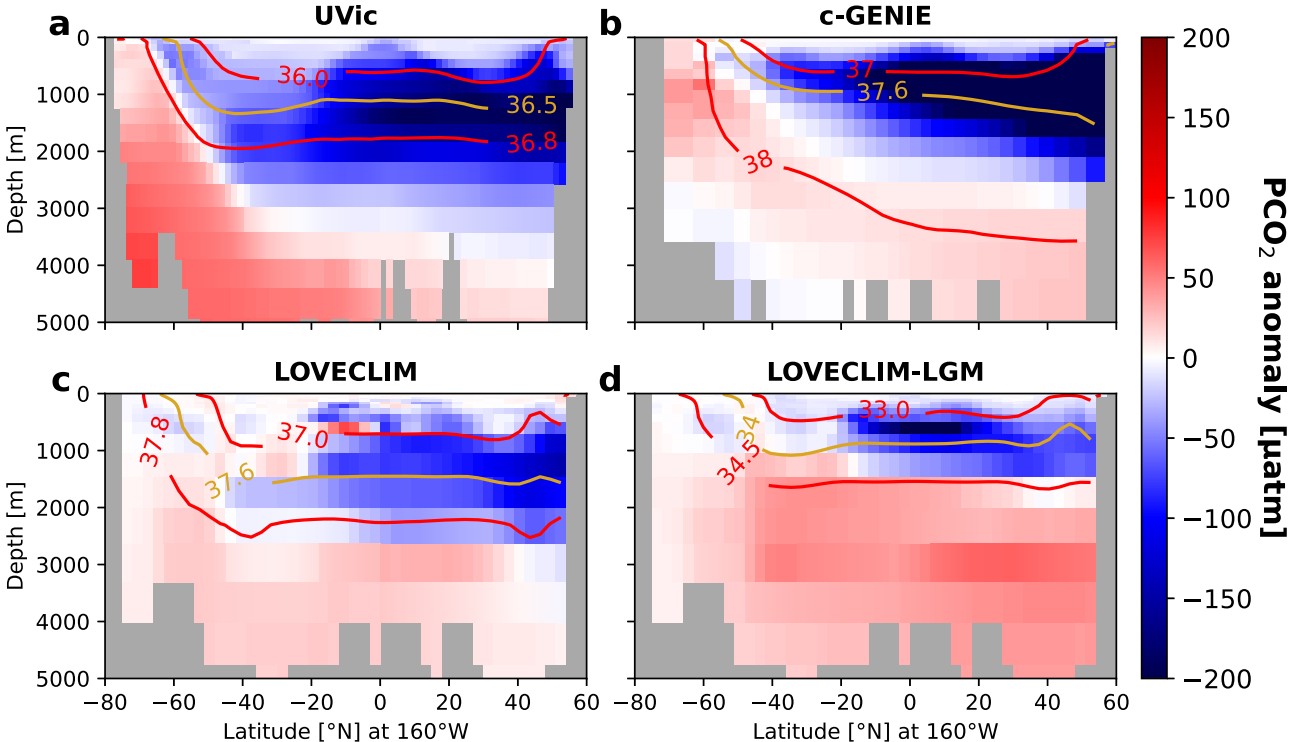

**Fig. 3 | Basin-wide reductions in sub-surface carbon (potential $p\mathrm{CO_2}$ or PCO₂, µatm) are a robust response to North Pacific ventilation across glacial-like Earth System Model simulations.** PCO₂ anomalies (µatm) along 160 °W resulting from a well-ventilated glacial North Pacific in the (**a**) UVic[53], (**b**) cGENIE[28], (**c, d**) and LOVECLIM[53,54] Earth System Models. PCO₂ was computed consistently across models (see "Methods"). PCO₂ denotes potential $p\mathrm{CO_2}$, where $p\mathrm{CO_2}$ is the partial pressure of carbon dioxide (see "Methods"). For models in (**a, c,** and **d**), a ventilated North Pacific was induced by North Atlantic freshwater input, enhancing North Pacific Intermediate Water (NPIW) formation. In (**b**), NPIW formation was induced by reducing the prescribed atmospheric freshwater flux from the North Atlantic to the North Pacific (see Supplementary Table 2). Beyond this, all differ slightly in resolution, configuration, and ventilation strength, establishing North Pacific ventilation as a robust and effective means of reducing sub-surface carbon (and nutrients, Supplementary Fig. 3) content in models. "LOVECLIM-LGM" (**d**) e.g., provides context of an estimate of LGM-like overturning[54] (Supplementary Table 1), though the actual magnitude of glacial Pacific overturning is not well constrained. These results should therefore be viewed as indicative rather than definitive. For a comparison of the models presented here, see Supplementary Table 2. Anomalies in each case refer to the perturbed (North Pacific-ventilated) simulation minus the control simulation. Anomalies compare averages over the last 10 years of each simulation, except in the case of c-GENIE, which compares the final year of each simulation. Red and yellow contours show isolines of σ₂ (potential density referenced to 2000 dbar, over the same time periods as the PCO₂ output) from the perturbed simulation in each case, bounding and characterizing the core of the low-PCO₂ water. These σ₂ isolines are meant to approximate the neutral surfaces along which these waters are expected to flow. Following convention, we subtract 1000 kg m⁻³ from all density values provided in this study. Note that we use σ₂ as a qualitative water mass marker, acknowledging that it does not delineate an exact neutral pathway of the water.

responses to such ventilation. The initial onset of overturning is expected to generate a short-lived transient peak in surface nutrients and carbon as convection initially taps into nutrient- and carbon-rich sub-surface waters below[56]. Over time, continued ventilation depletes this sub-surface reservoir, eventually leading to persistently low surface concentrations[28]. Thus, the long-term equilibrated response to ventilation of the North Pacific is one of persistently low carbon and nutrient concentrations in surface waters, even in the face of sustained convective mixing. The difference between the transient and equilibrated response is also well illustrated in ventilation experiments performed in cGENIE, which show the transient peak in surface nutrients subsiding within the first several hundred years of simulation (see Supplementary Fig. S9 in ref. 28). The same response is observed in the UVic simulations analyzed in this study, where North Pacific ventilation is triggered by North Atlantic freshwater perturbations under an otherwise glacial-like state. Despite the initial mixing up of deep carbon, ventilation reduces outgassing rates in the North Pacific within the first few centuries of the simulation, with strong reductions having developed by the end of 1000 years (Supplementary Fig. 5).

More active NPIW formation and ventilation thus provides a means of reducing surface carbon and nutrients in the glacial North Pacific by reducing the carbon and nutrient load of the subsurface waters that supply them. We propose that these North Pacific changes could also account for the reduced carbon and nutrient load in the glacial Southern Ocean surface. With better ventilation by an expanded "glacial NPIW" (GNPIW) cell, the previously carbon-rich Pacific mid-depths feeding the Southern Ocean would be replaced by carbon-/nutrient-poor waters from the surface. If this well-ventilated low-carbon/nutrient signal in the sub-surface glacial North Pacific reached the Southern Ocean via isopycnal diffusion as it does today[23], this would provide an alternative means of reducing the glacial Southern Ocean surface carbon and nutrient load (that could act independently of, or in conjunction with, physical changes like reduced upwelling or mixing), while simultaneously explaining the shared "Polar Twins" proxy pattern between the North Pacific and Southern Ocean. Indeed, this low-carbon low-nutrient signal is seen to propagate southward from the North Pacific in all models, notably along isopycnals outcropping in the Southern Ocean (red and yellow contours in Fig. 3 and Supplementary Fig. 3). This is a consistent pattern persisting across a range of Earth System Models spanning differing configurations, boundary conditions, and ventilation strengths, and thus is an effective demonstration of the ability of a well-ventilated North Pacific to reduce the carbon and nutrient content of waters feeding the Southern Ocean.

## Reduced carbon and nutrient delivery to the Southern Ocean

We now quantify the impact of a well-ventilated North Pacific on Southern Ocean carbon and nutrient content in the intermediate-complexity University of Victoria Earth System Climate Model (UVic ESCM) v2.9[57]. While mid-depth carbon/nutrient reduction is a robust response to North Pacific ventilation across the models, independent of model choice or even ventilation strength (Fig. 3, see also Chikamoto et al., 2012)[55], we chose this model to analyze in detail as it has the highest resolution (1.8° latitude x 3° longitude) compared to the other models presented in Fig. 3 (LOVECLIM/LOVECLIM-LGM: 3° x 3°; c-GENIE: 5° latitude x 10° longitude). Perhaps linked to this, UVic also most clearly showcases the signal of the dynamics we set out to demonstrate, with the low-carbon, low-nutrient anomaly propagating clearly from the North Pacific and being upwelled in the Southern Ocean. Thus, rather than a comprehensive and definitive representation of LGM overturning and biogeochemistry, we present these results as a test and proof of concept of an unexplored mechanism, one which we hope motivates future study in more targeted sets of model experiments.

The UVic simulation analyzed here was first presented by Menviel et al. (2014), who performed North Atlantic meltwater experiments in the UVic v2.9 and LOVECLIM v1.1 ESMs to investigate marine carbon cycle responses to Atlantic meridional overturning circulation (AMOC) shutdown[53]. In the North Atlantic meltwater simulations, 0.1 Sv of freshwater was added over the North Atlantic (50–65 °N and 55-10 °W) for 1000 years. After the 1000-year integration, the model has effectively reached equilibrium with respect to its biogeochemistry, with minimal residual trends in its atmosphere (< 0.3ppm/100 yr) and land and ocean (<1PgC/100 yr) carbon inventories. Conditions after 1000 years are therefore taken to be highly representative of a fully equilibrated state. In both UVic and LOVECLIM, this induced stronger North Pacific Intermediate Water (NPIW) formation (arising from a weaker North Pacific halocline resulting from oceanic and atmospheric teleconnections) and thus stronger ventilation of North Pacific waters. In UVic, this results in better ventilation down to about 2000 m depth and an expanded "GNPIW" ("glacial NPIW") overturning cell of ~14.8 Sv (Supplementary Fig. 2 and Supplementary Table 1)[53]. We therefore take advantage of this ventilated UVic simulation to study the impacts of a well-ventilated North Pacific on basin-wide biogeochemistry. We acknowledge that North Atlantic-North Pacific teleconnections are a topical area of research, with the strength of North Pacific ventilation and overturning resulting from North Atlantic freshwater perturbations having been demonstrated to differ across models (e.g., Saenko et al., 2004; Chikamoto et al., 2012; Baker et al., 2025)[55,58,59]. Enhanced North Pacific ventilation resulting from North Atlantic freshwater perturbations, however, remains a robust feature across models (including the simulations of Chikamoto et al., 2012, and Baker et al., 2025), and the details of its establishment do not impinge upon our results. Instead, we focus on the downstream impacts on basin-wide and Southern Ocean biogeochemistry of a ventilated North Pacific, for which there is glacial proxy evidence[51,52]. Furthermore, strong reductions in mid-depth carbon and nutrients appear to be a robust feature regardless of the magnitude of increase in ventilation strength (Chikamoto et al., 2012)[55], further supporting the ideas motivating this study.

We furthermore do not take the meltwater simulation (given its AMOC shutdown) as a precise analog of LGM conditions, but rather we capitalize on its representation of North Pacific ventilation and use it to assess the impacts of a well-ventilated North Pacific on Southern Ocean biogeochemistry. We do note, however, that similar results to those presented here are seen in simulations with LGM-like changes in overturning[54], with less extreme AMOC reduction and NPIW formation (Fig. 3d, Supplementary Fig. 2 and Supplementary Table 1, and Rae et al. (2020)[28]).

Here, we compare Southern Ocean carbon and nutrient content in two simulations of UVic, both run under identical LGM-like boundary conditions (Supplementary Table 2). One simulation features strong North Pacific ventilation driven by North Atlantic meltwater addition and associated enhanced NPIW formation (the perturbed simulation, hereafter referred to as "UVic-NP"), while the other serves as a control simulation, lacking meltwater addition and associated North Pacific ventilation ("UVic-ctrl") (compare overturning strength and radiocarbon age anomaly between UVic-NP and UVic-ctrl as an indicator of this ventilation, Supplementary Figs. 2 and 6, Supplementary Table 1). In these simulations we quantify carbon content by potential $pCO_2$ (PCO$_2$), which being a function of both alkalinity and dissolved inorganic carbon (DIC) is an effective measure of a water parcel's outgassing potential[12]. Nutrient content is traced through phosphate concentrations ([PO$_4^{3-}$]). We also quantify differences in Southern Ocean outgassing rates between the two simulations.

In the UVic-NP simulation, a stronger GNPIW overturning cell ventilates and reduces the carbon and nutrient content of sub-surface North Pacific waters – an anomaly which extends southwards through the basin along mid-depths (Fig. 3a and Supplementary Fig. 3a). We see strong reductions in PCO$_2$ in the North Pacific from the near surface down to ~3000 m depth, with a mean PCO$_2$ anomaly of −199 μatm in the sub-surface North Pacific (30–60 °N, 140–240 °E, 500–2000 m) (Fig. 4a). Phosphate concentration is similarly reduced (Supplementary Fig. 3a and Fig. 4a), with a mean anomaly of −1.3 μmol kg$^{-1}$ in the sub-surface North Pacific. The low-PCO$_2$ signal extends along isopycnals through mid-depths (~1000–2000 m) to southwards of 30 °S, exhibiting a concentrated core (−78 μatm on average) in the western sector of the basin (140–200-°E, 1000–2500 m), while more diffuse anomalies of −47 μatm on average pervade the basin (140–280 °E, 1000–2500 m) (Fig. 4a, c). After ~50°S, this low-PCO$_2$ signal is translated upwards along isopycnal surfaces (σ$_2$ = 36.0-36.8 kg m$^{-3}$), outcropping in the Southern Ocean surface (red and yellow contours in Fig. 3a). Again, similar patterns are seen in phosphate concentrations (Fig. 4d), where at 30 °S phosphate is reduced by −0.3 μmol kg$^{-1}$ on average across the basin and by −0.5 μmol kg$^{-1}$ in the western sector of the basin. This negative phosphate anomaly similarly reaches the Southern Ocean surface along outcropping isopycnals (Supplementary Fig. 3a).

## Low PCO$_2$ drives reduced Southern Ocean outgassing

This low-PCO$_2$ low-nutrient signal propagated from the North Pacific translates into quantifiable negative PCO$_2$ and phosphate anomalies throughout the Southern Ocean surface, spanning the Subantarctic and Antarctic Zones (Fig. 5a, b). We find the strongest anomalies in the Pacific-sector (150–280 °E) Subantarctic Zone (SAZ, ~40–60 °S), where mid-depth Pacific isopycnals bounding this low-PCO$_2$ water (σ$_2$ = 36.0-36.8 kg m$^{-3}$, red contours in Fig. 5) outcrop. We observe a mean PCO$_2$ anomaly of approximately −8 μatm across the Southern Ocean (all longitudes south of 40 °S) and of approximately −12 μatm across the Indian-Pacific sector (25–280 °E), with maximum anomalies surpassing approximately −25 μatm (Fig. 5b). Phosphate is reduced by −0.13 μmol kg$^{-1}$ on average across the Southern Ocean and by −0.17 μmol kg$^{-1}$ in the Indian-Pacific sector, with maximum anomalies surpassing −0.44 μmol kg$^{-1}$ (Fig. 5a).

We also note that ventilation could likely induce even greater carbon anomalies than documented here if applied to a system transitioning out of interglacial conditions. Having been run under glacial-like conditions, UVic-ctrl exhibits a low-carbon bias in the North Pacific (PCO$_2$ ~= 350–450 μatm) relative to a pre-industrial or interglacial-like state (>1000 μatm[24], Fig. 1). Starting from a more carbon-rich baseline in UVic-ctrl could have enabled an even greater reduction in carbon and nutrients at mid-depths of the North Pacific in the perturbed simulation than we observe here, with downstream effects on the Southern Ocean. Thus, in terms of the magnitude of carbon drawdown

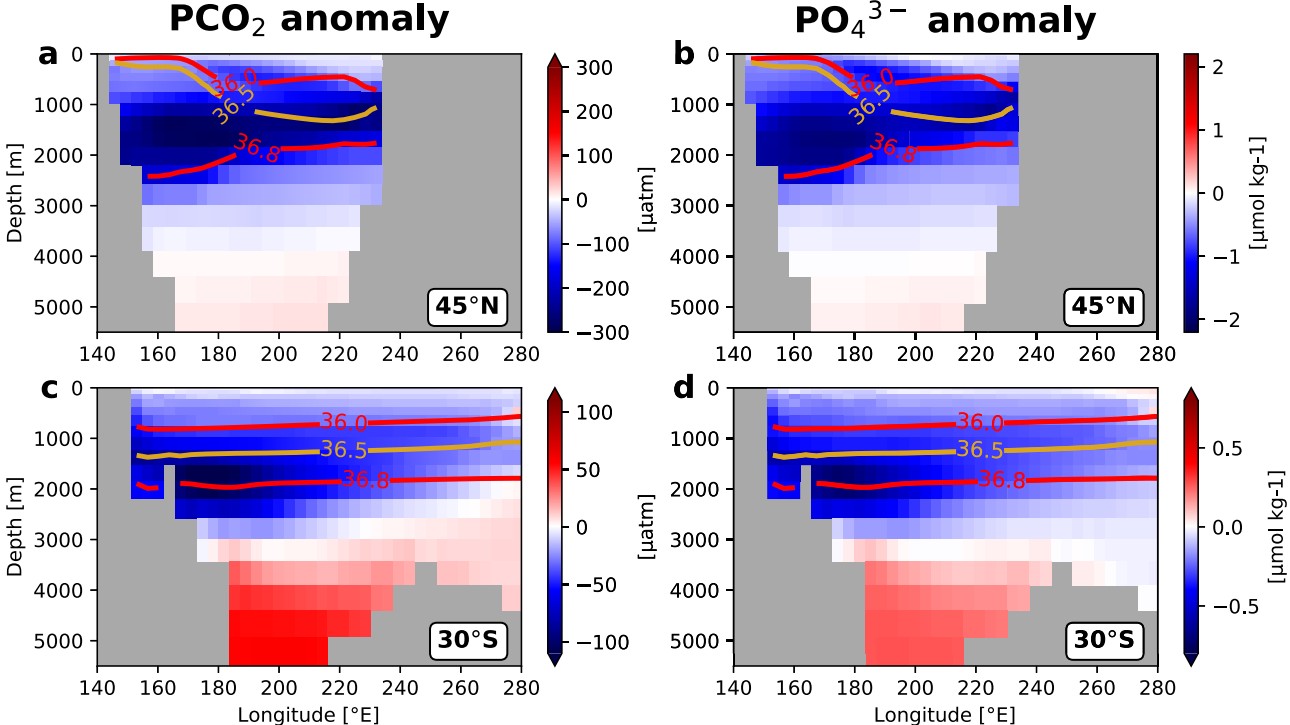

**Fig. 4 | Negative potential $pCO_2$ (PCO$_2$) [μatm] and phosphate (PO$_4^{3-}$) (μmol kg$^{-1}$) anomalies resulting from North Pacific ventilation pervade across the Pacific basin.** PCO$_2$ anomaly at (**a**) 45 °N and (**c**) 30 °S across the Pacific basin, and the same for (**b**, **d**) phosphate anomaly. PCO$_2$ denotes potential $pCO_2$, where $pCO_2$ is the partial pressure of carbon dioxide (see "Methods"). Anomalies refer to the perturbed simulation minus the control simulation (UVic-NP − UVic-ctrl) and compare averages over the last 10 years of each simulation. Red and yellow contours show isolines of σ$_2$ (potential density referenced to 2000 dbar) from the perturbed simulation (averaged over the last 10 years of the UVic-NP simulation), meant to approximate the neutral surfaces along which these waters are expected to flow.

associated with ventilation change, our results may represent a conservative estimate.

Even so, the negative PCO$_2$ anomalies at the surface of the Southern Ocean reduce CO$_2$ outgassing, chiefly in the Subantarctic Zone (~40–60 °S) (Fig. 5c). This translates into a mean Southern Ocean anomaly (in all areas south of 40 °S) of − 0.18 mol m$^{-2}$ yr$^{-1}$ (representing a ~50% reduction relative to UVic-ctrl's mean outgassing rate). The Indian-Pacific sector (25–280 °E, south of 40 °S) sees a mean anomaly of − 0.26 mol m$^{-2}$ yr$^{-1}$ or ~79% relative to the control. Furthermore, reduced outgassing is demonstrably the result of the altered biogeochemistry of the upwelling water. Changes in alkalinity and DIC dominate the outgassing anomaly signal (Fig. 5d), whereas temperature and salinity changes play a relatively minor role (see Supplementary Fig. 7). The density surfaces bounding the core low-PCO$_2$ water mass from the North Pacific (σ$_2$ = 36.0–36.8 kg m$^{-3}$, red contours in Fig. 5) outcrop at the Southern Ocean surface and delineate the negative air-sea CO$_2$ flux anomaly, further supporting our proposed North Pacific-based mechanism.

With such major circulation changes as a weakened AMOC and a well-ventilated North Pacific in this simulation, changes in Southern Ocean circulation dynamics may also be expected. Thus, in order to link the reduced outgassing signal seen in the Southern Ocean chiefly to the reduced carbon content of the upwelled waters, two other potential processes must be ruled out: reduced upwelling and/or enhanced stratification (reducing mixing). As UVic's atmosphere is simulated using a non-dynamic energy-moisture balance model (in which the equations for momentum conservation are replaced with climatological wind data and only minor dynamical feedbacks alter this climatology), virtually no change (< 0.5%) in the mean wind stress curl over the high-latitude (south of 50 °S) Southern Ocean is found between the control and perturbed simulations. Reduced wind-driven

upwelling is therefore unlikely to explain the reduced Southern Ocean outgassing. Stratification (quantified as the density gradient between 500 m and the surface) similarly only increases by approximately + 1.5% on average across the Southern Ocean (south of 40 °S) from UVic-ctrl to UVic-NP, and so the reduced outgassing is also unlikely to be the result of reduced mixing isolating surface waters from carbon-rich deep waters below. Instead, it is clear that the ability of biology to "keep pace" with the incoming carbon/nutrient supply has been improved; with physical supply via mixing or upwelling being virtually unchanged, the reduced carbon/nutrient content of the supplying water plays the central role.

An additional factor potentially impacting outgassing is biological productivity. If productivity were significantly enhanced in UVic-NP, this could increase carbon export from surface waters, potentially contributing to CO$_2$ undersaturation and reduced outgassing in our output. Instead, we find net primary production (Supplementary Fig. 8) is slightly reduced in our experiment (by ~19% in the Pacific sector (150–280 °E) of the Southern Ocean and by ~3% on average across the whole Southern Ocean, south of 50 °S), consistent with the observed reduction in surface nutrients (Fig. 5a). Reduced biological productivity would, in isolation, act to enhance outgassing by reducing the uptake of carbon into organic matter. However, the simulated reduction in outgassing alongside reduced levels of productivity highlights the significant role that the water's biogeochemical composition and carbon content play in dictating its outgassing potential. Again, even with overall productivity being slightly reduced, the ability of biology to keep pace with its carbon/nutrient supply and counteract CO$_2$ evasion is clearly improved, evident by reduced overall outgassing.

Finally, other factors besides ocean carbon content influencing outgassing fluxes include wind speed, sea ice cover, and atmospheric

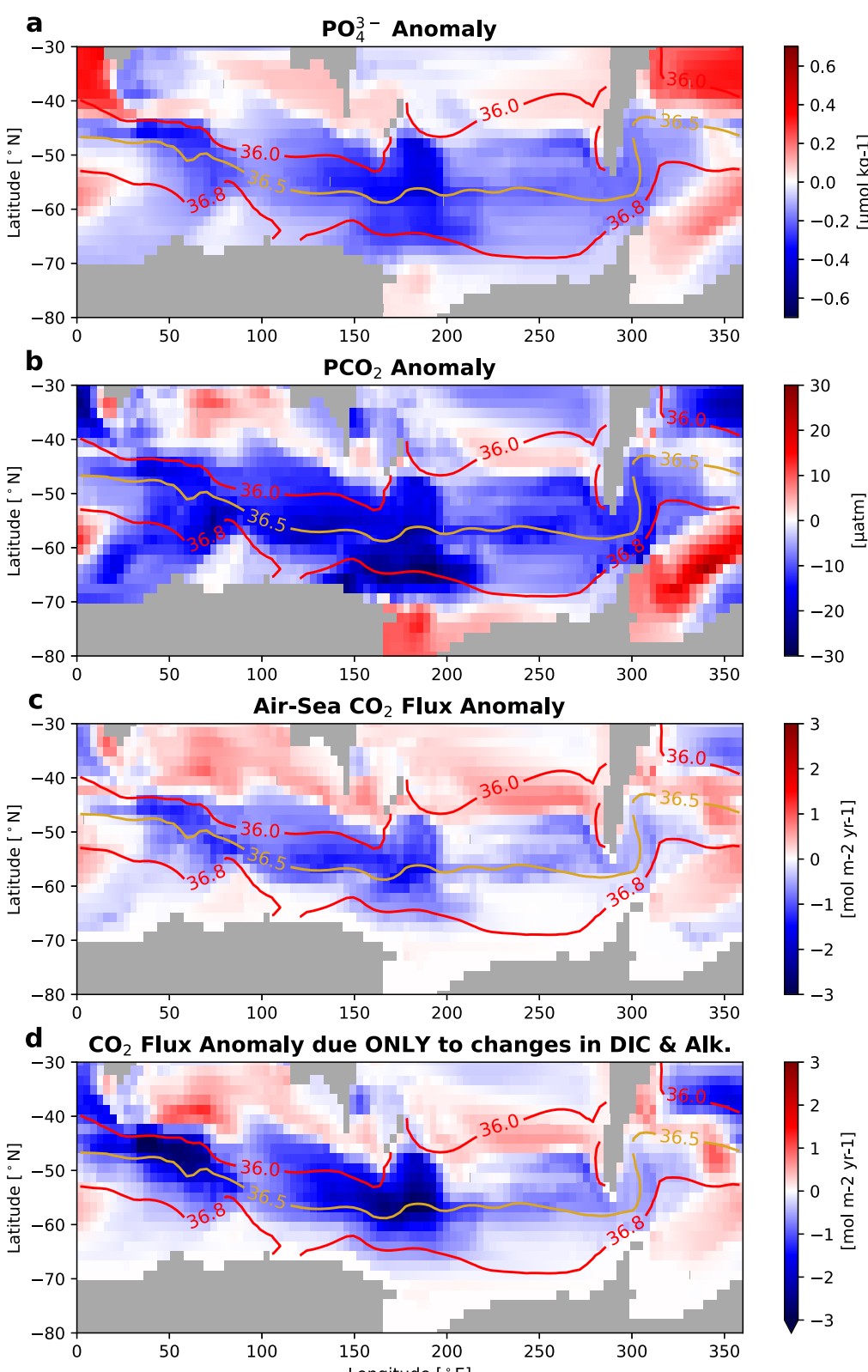

pCO₂. Ocean-to-atmosphere flux of CO₂ [mol m⁻² sec⁻¹] is calculated in the model[57,60] as

$$f = k_w * [C_{sw} - C_{atm}] \quad (1)$$

where $k_w$ is the gas transfer velocity [m sec⁻¹] and $C_{sw}$ and $C_{atm}$ represent aqueous CO₂ concentration [mol m⁻³] (see "Methods" for

details). Changes in atmospheric pCO₂ (impacting $C_{atm}$) as well as wind speed and fractional sea ice cover (impacting $k_w$) must therefore also be accounted for to attribute the reduced outgassing signal to the reduced carbon content of the water. Supplementary Fig. 7 decomposes the total flux anomaly (UVic-NP − UVic-ctrl) into the respective contributions from each of these factors. To determine the magnitude of flux anomaly attributable to a specific factor (e.g., change in wind

**Fig. 5 | The Southern Ocean surface exhibits negative anomalies in phosphate ($PO_4^{-3}$) (µmol kg$^{-1}$), potential $p$CO$_2$ (PCO$_2$) (µatm), and air-sea CO$_2$ flux (mol m$^{-2}$ yr$^{-1}$) as a result of North Pacific ventilation.** Anomalies resulting from a well-ventilated North Pacific (i.e., UVic-NP – UVic-ctrl) in Southern Ocean surface (**a**) phosphate (µmol kg$^{-1}$), (**b**) PCO$_2$ (µatm), (**c**) air-sea CO$_2$ flux (mol m$^{-2}$ yr$^{-1}$), and (**d**) air-sea CO$_2$ flux (mol m$^{-2}$ yr$^{-1}$) due only to changes in the water's alkalinity and dissolved inorganic carbon (DIC) between the simulations. PCO$_2$ denotes potential $p$CO$_2$, where $p$CO$_2$ is the partial pressure of carbon dioxide (see "Methods"). Negative anomalies in air-sea flux indicate reduced CO$_2$ outgassing or enhanced CO$_2$ uptake. Anomalies compare averages over the last 10 years of each simulation. In (**d**), air-sea flux is first calculated in the UVic-NP simulation, and flux anomaly is then calculated as the difference between using control (UVic-ctrl) alkalinity and DIC (with all other variables held at UVic-NP values) and perturbed-simulation (UVic-NP) alkalinity and DIC, thus isolating the flux anomaly signal due only to seawater chemistry changes. Red and yellow contours show isolines of σ$_2$ (potential density referenced to 2000 dbar) from the perturbed simulation (averaged over the last 10 years of the UVic-NP simulation). σ$_2$ is plotted at 500 m depth here to avoid the influence of seasonal mixed layer processes.

speed between UVic-NP and UVic-ctrl), we compute flux using the factor of interest as simulated the UVic-NP simulation, while using fields for all other variables from the UVic-ctrl simulation. The resulting flux field then has subtracted from it the flux field obtained from the UVic-ctrl simulation, computed using UVic-ctrl fields for all variables. The resulting anomaly thus gives the anomaly in flux due only to changes in the factor of interest between UVic-ctrl and UVic-NP.

Winds and sea ice cover are virtually unchanged between the control and perturbed simulations, resulting in negligible contributions to the reduced outgassing signal (Supplementary Fig. 7e, f). Atmospheric $p$CO$_2$ drops slightly from 191 µatm in UVic-ctrl to 185 µatm in UVic-NP. In isolation, this reduced atmospheric $p$CO$_2$ would drive greater outgassing (Supplementary Fig. 7d). The influence of this atmospheric $p$CO$_2$ change thus slightly dampens the negative outgassing anomaly in the Southern Ocean obtained in the perturbed simulation (Supplementary Fig. 7b). It is therefore changes in the water's biogeochemistry, resulting from a better ventilated North Pacific in UVic-NP, that account for the reduced Southern Ocean outgassing.

## Discussion

Poorly-ventilated mid-depth waters from the North Pacific represent a major source of carbon and nutrients to the Southern Ocean surface today[12,13]. Southern Ocean outgassing is concentrated in a narrow ring between the wintertime sea ice edge and the Subantarctic Front that coincides with deep mixed layers intermittently reaching the carbon-rich waters of Indian-Pacific origin straddling the 27.8 kg m$^{-3}$ neutral density surface – a pathway referred to as the "deep ocean's carbon exhaust"[12]. Reducing the carbon content of these mid-depth Pacific waters – via better ventilation by enhanced GNPIW formation as emerges in our simulations – thus provides an effective way of reducing Southern Ocean outgassing. This result holds significant implications for theories of glacial CO$_2$ change based on stemming Southern Ocean outgassing, demonstrating that a central feature of these theories and of proxy data – a reduced glacial carbon and nutrient load in the Southern Ocean surface – may have been strongly influenced by improved ventilation of the North Pacific and associated changes to the biogeochemistry of the waters feeding the Southern Ocean.

The dilution of the North Pacific's mid-depth carbon reservoir is not inconsistent with a scenario of glacial CO$_2$ drawdown. While we expect some of the carbon lost from these mid-depths to have initially been outgassed to the atmosphere (both locally and at lower near-equatorial latitudes)[61], this represents only the transient response to North Pacific ventilation[28,56]. This short-lived response has been demonstrated in previous studies designed to simulate transient events (e.g., Heinrich events), which show a transient increase in atmospheric CO$_2$ following the onset of North Pacific ventilation (e.g., Menviel et al., 2012, 2014; Rae et al. 2014)[53,56,61]. By contrast, we probe the present simulations for their implications for the long-term equilibration response to ventilation, relevant to sustained glacial conditions. Despite an initial release of carbon to North Pacific surface waters following ventilation (as described by previous studies), we find that North Pacific outgassing ultimately (after 1000 years) decreases in

our simulations (Supplementary Fig. 5). Indeed, outgassing is reduced in response to ventilation across three key outgassing regions of the Indian-Pacific basin: the North Pacific, the Eastern Equatorial Pacific, and the Southern Ocean (Supplementary Fig. 9). These regions in the control state likely see net outgassing as a result of biological carbon fixation being outpaced by an over-supply of carbon and nutrients, which North Pacific ventilation alleviates by reducing the carbon/nutrient load of the supplying sub-surface waters. By contrast, positive anomalies in outgassing are only seen in subtropical latitudes (Supplementary Fig. 9). The key difference here is that nutrients supplied to these regions ultimately stay in surface waters, fated to drive compensatory biological carbon fixation elsewhere. Nutrients upwelled in the Southern Ocean, in contrast, if not used by biology, can be circulated back down into the abyss and lose their compensatory potential. For this reason, the transfer of outgassing anomalies from the Southern Ocean to subtropical latitudes is crucial for drawing down atmospheric CO$_2$.

While we cannot precisely quantify the ocean's long-term carbon uptake response to North Pacific ventilation in the present simulation (due to the confounding effects of the freshwater forcing weakening the AMOC and inducing outgassing in the North Atlantic, an effect unrelated to North Pacific ventilation – see the caption of Supplementary Fig. 5), we do observe a ~6 µatm decrease in atmospheric $p$CO$_2$ from UVic-ctrl to UVic-NP, reflecting the accumulation of carbon in the ocean (the land carbon inventory by contrast slightly decreases in UVic-NP (not shown) and thus cannot account for this change in the atmosphere). Consistent with this, our results point to a significant shift of carbon out of mid-depths and into the deep ocean (Fig. 3), akin to the "nutrient-deepening" hypothesis of Boyle (1988)[62]. This finding highlights the need for further investigation of the North Pacific's influence on Southern Ocean biogeochemistry, and future work should focus on quantifying changes to the ocean carbon budget in response to North Pacific ventilation. We furthermore present this as a contributing mechanism to glacial CO$_2$ change, not expecting it to account for the full magnitude of glacial change but operating alongside other processes well-evidenced by proxies and modeling studies such as iron fertilization[18–21], changes in sea ice and disequilibrium carbon[63–65], and shifts in bottom water properties and stratification[66–69].

Alongside its implications for outgassing and glacial CO$_2$ change, our results offer an as of yet unconsidered explanation for the shared "Polar Twins" pattern between the North Pacific and Southern Ocean of a reduced glacial nutrient supply. Previous theories have attributed this pattern to the simultaneous action of certain physical processes in each region, such as enhanced upper-ocean stratification (< 500 m) limiting the mixing of sub-surface carbon and nutrients to the surface[26]. While a reduced nutrient supply is well-supported in both regions in proxy data[35,41,43–48,51,52] (Fig. 2), this pattern is not exclusive to enhanced stratification, and proxy data in fact support enhanced ventilation and convection in the North Pacific at the LGM[28]. In the Southern Ocean, while multiple proxy and modeling studies support enhanced glacial stratification at depth (e.g., Burke and Robinson, 2012; Skinner et al., 2010; Roberts et al., 2016)[70–72], near-surface stratification is found to be reduced under glacial conditions in a variety of models, for instance, due to enhanced brine rejection[68,73–75].

Alternative mechanisms to explain the Polar Twins proxy pattern of reduced carbon/nutrient supply are thus useful, given that at least some glacial conditions in the Southern Ocean (e.g., increased sea ice formation and brine rejection) are expected to oppose stratification. Under such conditions, a reduction of the carbon and nutrient content of supplying sub-surface waters would then become essential to reducing the carbon and nutrient supply to the Southern Ocean surface.

A reduction in wind-driven upwelling has also been suggested to isolate glacial Southern Ocean and North Pacific surface waters from carbon- and nutrient-rich deep waters below[27]. However, PMIP3 models suggest that wind-driven upwelling was actually enhanced in the glacial North Pacific[49]. Winds over the glacial Southern Ocean, by contrast, may have been weaker[76]. A recent analysis combining sediment proxies and climate model output indicates a ~25% weakening and 5° equatorward shift of the southern westerlies at the LGM[76]. This weakening likely reduced the upwelling of carbon-rich deep waters, stemming surface outgassing. However, the equatorward shift would also have positioned the winds to preferentially pull up lighter isopycnals – including the mid-depth Pacific waters highlighted in this study that are today rich in carbon (Fig. 1). Thus, while weakened Southern Ocean upwelling may have reduced the surface carbon supply and contributed to $CO_2$ drawdown, a reduction in the carbon content of these mid-depth Pacific waters could have provided an additional crucial contribution. Furthermore, given the potential of such wind-driven Southern Ocean upwelling to drive Pacific overturning[59], the glacial equatorward shift in winds may have strengthened the physical and biogeochemical coupling between the North Pacific and Southern Ocean. Our results thus bring a fresh perspective to the Polar Twins pattern, demonstrating how the carbon and nutrient content of both the Southern Ocean and North Pacific surface could have been altered by North Pacific ventilation.

We have explored how a better-ventilated glacial North Pacific influences Southern Ocean biogeochemistry. We have demonstrated that stronger ventilation of the North Pacific reduces the carbon and nutrient load of mid-depth waters feeding into the Southern Ocean, leading directly to reduced surface carbon and nutrient load and reduced Subantarctic Zone outgassing. This result provides an as of yet unconsidered explanation for the shared "Polar Twins" proxy pattern between the glacial North Pacific and Southern Ocean. While not ruling out possible changes to upwelling or mixing, our findings demonstrate the ability of reduced carbon- and nutrient-content of water – outside of any reduction in physical delivery to the surface – to significantly alter Southern Ocean surface biogeochemistry. A reduced carbon and nutrient load of the Southern Ocean surface, given its impact on outgassing, is notably a potentially important mechanism in explaining glacial $CO_2$ change. In this context, our results highlight the importance of source-water biogeochemistry alongside local physical pathways and processes in theories of glacial $CO_2$ change that invoke stemming the $CO_2$ leak from the Southern Ocean. They also underscore the importance of north-south connectivity along Pacific mid-depths[12,23] and associated interhemispheric ventilation dynamics.

Given its ability to explain key proxy records and its influence on Southern Ocean outgassing demonstrated here, North Pacific ventilation emerges as an important feature of the glacial ocean, warranting further study. More dedicated consideration of this feature could offer insight into model–data mismatches and help guide the direction of future glacial modeling efforts. As such, this will be an important consideration for future studies as we continue to unravel mechanisms of ocean carbon uptake across glacial, contemporary, and future time scales.

## Methods
### UVic Earth system model simulations
The UVic ESM uses the Modular Ocean Model (MOM) version 2, a three-dimensional ocean general circulation model with a 3.6° longitude x 1.8° latitude horizontal resolution and 19 depth layers[57,77]. The UVic ESM also includes a vertically integrated two-dimensional energy-moisture balance model of the atmosphere, a dynamic-thermodynamic sea ice model, a land surface scheme, a dynamic global vegetation model, a marine carbon cycle model, and a sediment model[57]. UVic uses the marine ecosystem/biogeochemical model of Schmittner et al. (2008)[78]. Marine carbon chemistry and air-sea gas exchange follow Ocean Carbon-Cycle Model Intercomparison Project (OCMIP) protocols[79]. Atmospheric carbon dioxide is fully coupled and radiatively active. A full inorganic carbon cycle is simulated as well as organic carbon components, with particulate carbon being remineralized into both dissolved inorganic carbon and dissolved organic matter pools. Biological productivity is co-limited by nitrate, phosphate, and irradiance, with a temperature-dependence on maximum phytoplankton and diazotroph growth rate[80].

To investigate the impact of a well-ventilated North Pacific on carbon delivery to the surface Southern Ocean, we compare a perturbed simulation in which strong ventilation occurs due to enhanced North Pacific Intermediate Water (NPIW) formation (hereafter referred to as "UVic-NP") with a control simulation run under identical conditions but lacking the North Atlantic meltwater addition and associated enhancement of NPIW formation ("UVic-ctrl"). The perturbed UVic-NP simulation was forced by a freshwater hosing of 0.1 Sv imposed on the North Atlantic (over a region spanning 50–65 °N and 55-10 °W) for 1000 years. Both the control and perturbed simulations were otherwise run under constant LGM-like (~21 ka BP) boundary conditions for orbital parameters and ice sheet topography and albedo; $CO_2$ was set to be prognostic in these simulations, though the LGM-like background state was first established by forcing the model under LGM $CO_2$ (192 ppm) until a quasi-equilibrium was reached[53]. The simulations were compared taking averages over the last 10 years of each simulation.

### PCO$_2$ Calculation
Following the example of Chen et al. (2022) and Prend et al. (2022), we trace carbon dynamics in the Pacific basin and Southern Ocean using potential $pCO_2$ (PCO$_2$), that is, the partial pressure of $CO_2$ ($pCO_2$) a water parcel would have if brought to the surface adiabatically (thus correcting for the effect of pressure on both temperature and partial pressure)[12,13]. This property, being a function of both dissolved inorganic carbon (DIC) and alkalinity, is of more direct relevance to a water parcel's outgassing or in-gassing potential than DIC alone; sub-surface waters of greater PCO$_2$ than atmospheric $pCO_2$ would outgas if brought to the surface. It is possible for waters of high DIC content to actually have low outgassing potential (low PCO$_2$) if buffered by high alkalinity or if exhibiting high $CO_2$ solubility (e.g., by way of lower water temperatures). PCO$_2$ thus represents a more accurate measure of outgassing potential than DIC alone.

PCO$_2$ is derived by calculating $pCO_2$ referenced to 0 dbar from sub-surface alkalinity, DIC, potential temperature, and salinity from the model output using PyCO2SYS v1.8.1[81]. The calculation uses the dissociation constants of carbonic acid from Dickson & Millero (1987)[82], of bisulphate from Dickson (1990)[83], and of hydrogen fluoride from Dickson & Riley (1979)[84], and the boron to salinity ratio of Lee et al. (2010)[85].

### CO$_2$ Flux Calculation
Flux of $CO_2$ [mol m$^{-2}$ sec$^{-1}$] from the ocean to the atmosphere is calculated in UVic[57,60] as

$$f = k_w * [C_{sw} - C_{atm}] \tag{2}$$

where $k_w$ is the gas transfer velocity [m sec$^{-1}$] and $C_{sw}$ and $C_{atm}$ represent aqueous $CO_2$ concentration [mol m$^{-3}$].

$C_{sw}$ [mol m$^{-3}$] is calculated from dissolved inorganic carbon, alkalinity, phosphate, temperature, and salinity in the surface depth level of UVic using PyCO2SYS v1.8.1[81]. The same dissociation constants are used as described in the "PCO$_2$ Calculation" section.

$C_{atm}$ [mol m$^{-3}$] is calculated from atmospheric $p$CO$_2$ from the model output and CO$_2$ solubility, calculated as a function of temperature and salinity[60]. Solubility ($ff$, in mol kg$^{-1}$ atm$^{-1}$) is calculated as

$$ff = \left( A_1 + A_2 \frac{100}{T} + A_3 \ln\left(\frac{T}{100}\right) + A_4 \left(\frac{T}{100}\right)^2 + S\left(B_1 + B_2 \frac{T}{100} + B_3 \left(\frac{T}{100}\right)^2\right) \right) \tag{3}$$

where $A_1 = -162.8301$, $A_2 = 218.2968$, $A_3 = 90.9241$, $A_4 = -1.47696$, $B_1 = 0.025695$, $B_2 = -0.025225$, $B_3 = 0.0049867$, $S$ = salinity in g kg$^{-1}$ or per mille, and $T$ = temperature in Kelvin. $ff$ is converted into mol m$^{-3}$ atm$^{-1}$ using the density of seawater, and from this $C_{atm}$ [mol m$^{-3}$] is then calculated as $C_{atm} = ff * p\text{CO}_{2,atm}$ where atmospheric $p$CO$_2$ is 185 µatm in the perturbed simulation ("UVic-NP") and 191 µatm in the control simulation ("UVic-ctrl").

The gas transfer velocity (or piston velocity) $k_w$ (in cm s$^{-1}$) is calculated as

$$k_w = (1 - \gamma_{seaice}) * 0.377 * \left(\mathbb{V}_s^2\right) * \left(\frac{Sc}{660}\right)^{-0.5} \tag{4}$$

where $\gamma_{seaice}$ is fractional sea ice cover (0-1), $\mathbb{V}_s$ is the average surface wind speeds (in m s$^{-1}$, described below), and $Sc$ is the Schmidt number (described below). This formulation gives $k_w$ in cm s$^{-1}$, and so it is first converted into m s$^{-1}$ before being used in Eq. (1)/(2). The 0.377 is a proportionality constant relating the gas transfer velocity to the average wind speed squared ($\mathbb{V}_s^2$)[60]. $\mathbb{V}_s$ is the average wind speed from the model output subjected to a 0.8 contraction factor and 20° rotation as described in the manuscript describing UVic[57]. The Schmidt number represents a further dependency of the gas transfer velocity on temperature and salinity and is calculated as described in the appendix of Wanninkhof (1992)[60].

## Data availability
The Earth System Model output used in this study is available in a Zenodo repository with https://zenodo.org/records/15276314[86].

## Code availability
Scripts used to analyze the Earth System Model output for this study are available in a GitHub repository with https://github.com/Maddie-Sh/ShankleEtAl2025_NPacSObgc[87].

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

## Acknowledgements

This work was supported by the following funding awards: a St Leonard's World-Leading Doctoral Scholarship from the University of St Andrews (M.G.S.), UKRI grant MR/W013835/1 (G.A.M.), support from the French national LEFE program through the ROOF project (W.R.G., C.dL.), Australian Research Council grant SR200100008 (L.C.M.), and NERC grant NE/N011716/1 (J.W.B.R., A.B.). This research was supported by the Australian Government's National Collaborative Research Infrastructure Strategy (NCRIS), with access to computational resources provided by the National Computational Infrastructure (NCI) through the National Computational Merit Allocation Scheme (L.M.).

## Author contributions

J.R., A.B., G.M. and M.S. designed the study. L.M. ran the Earth System Model simulations, which M.S. analyzed. M.S. also compiled all proxy data and wrote the manuscript. All authors (M.S., G.M., W.G., C.dL., L.M., A.B. and J.R.) contributed to the interpretation of the results and commented on the writing of the manuscript.

## Competing interests

The authors declare no competing interests.
