## [Transparent Peer Review file · Nature Communications]

Southern Ocean CO₂ outgassing and nutrient load reduced by a well-ventilated glacial North Pacific

Corresponding Author: Dr Madison Shankle

Version 0:

Reviewer comments:

Reviewer #1

(Remarks to the Author)

General Evaluation:

Summary and Strengths of the Study:

This study investigates the mechanisms behind low atmospheric CO₂ levels during glacial periods by analyzing the relationship between the Southern Ocean and the North Pacific intermediate waters using both models and proxy data. The authors propose that the expansion of North Pacific intermediate water (NPIW) during glacial periods may have led to a decrease in carbon and nutrient concentrations in the surface waters of the Southern Ocean, thereby suppressing CO₂ release. This perspective extends beyond the commonly discussed mechanisms of Southern Ocean stratification and westerly wind shifts, introducing a remote influence via intermediate waters.

A key strength of the study is the use of multiple models to examine the response of NPIW to freshwater input in the North Atlantic. This approach allows for a robust assessment of how North Pacific intermediate circulation could contribute to global carbon cycle changes. Additionally, the study is well-situated within the existing literature, particularly in its focus on how changes in the intrinsic carbon and nutrient concentrations of water masses upwelling in the Southern Ocean may have played a crucial role. The motivation and research objectives are clearly defined.

Areas for Improvement:

Conceptual Clarity and Positioning:

The study does not clearly define the specific region of the Southern Ocean being examined. Based on the text, it appears that the focus is on the subantarctic zone (approximately 45°S–55°S). Given that previous proxy-based studies (e.g., Sigman et al., 2021) have discussed regional differences in biological production and nutrient consumption efficiency between the subantarctic and Antarctic zones, the study should clarify which region its conclusions apply to. If the analysis aims to track changes along neutral density surfaces of approximately 27.5–28 kg m⁻³ in the modern ocean, this should be explicitly stated.

The response of North Pacific intermediate ventilation to North Atlantic freshwater forcing appears to differ between models. Some models show little strengthening of NPIW in response to freshwater input into the North Atlantic (e.g., Chikamoto et al., 2012). The study finds that MIROC simulations exhibit a decrease in nutrient concentrations from NPIW to the Southern Ocean, attributed to a reduction in the nutrient content of waters originating from the south. This interpretation differs from this study. To better support the conclusions regarding the role of NPIW in altering nutrient supply, a more thorough comparison with past studies is necessary. The current citations and discussions on this aspect are insufficient.

To strengthen the interpretation of ocean circulation timescale changes, it would be helpful to present $\Delta^{14}\text{C}$ or ideal age tracers as indicators of ventilation time. This would enable comparisons with sediment records and provide a useful benchmark for evaluating model responses.

Methodology:

In the paragraph starting at L290 and in Supplementary Figure 4, the study attempts to disentangle the factors contributing to the pCO₂ anomaly. However, the method used to separate the contributions of sea ice and wind speed is not sufficiently explained. Providing more details on this methodology would enhance clarity.

Results and Interpretation

Supplementary Figure 5 shows that when NPIW strengthens, the immediate response includes an increase in surface nutrient and carbon concentrations. Would this increase in nutrients not contribute to higher pCO₂ in low-latitude surface waters, potentially offsetting the negative pCO₂ anomaly in the Southern Ocean? This point needs to be examined further to ensure that the proposed mechanism effectively explains the observed CO₂ drawdown.

Although the study demonstrates that Southern Ocean pCO₂ decreases, low-latitude and North Atlantic surface pCO₂ increases (Supplementary Figure 3). This raises the question of whether a strengthened NPIW transported CO₂-rich waters from the North Pacific and subantarctic regions to other ocean basins via surface pathways. To strengthen this argument, the study should quantify global-scale surface ocean pCO₂ changes.

The study contributes to the “Polar Twins” framework, but it remains unclear whether the authors consider changes in NPIW alone to be sufficient in explaining the large carbon reservoir in the Southern Ocean. Alternatively, are additional processes, such as deep circulation changes and stratification, still necessary? Southern Ocean deep waters are characterized by high salinity, low δ¹³C, and depleted Δ¹⁴C in proxy records—can these be explained without invoking stratification? Addressing this point would clarify the broader implications of the study’s findings.

Presentation and Writing:

The figure readability should be improved. Axis labels and contour labels should be enlarged for better visibility.

Figure 2: Additional explanations should be provided on what the proxy data represent (e.g., whether higher/lower values indicate increased or decreased biological productivity or nutrient depletion).

Figures 3 and 4: Contour line colors and label sizes should be adjusted to enhance visibility.

Specific Comments:

L67: “Today the Southern Ocean receives most of its carbon and nutrients from mid- depths (1-3 km) of the Pacific basin⁹, where poor ventilation allows for their accumulation¹⁵.” Which ocean region does the Southern Ocean refer to? Also, does “Most” imply that nutrient supply comes more from the Pacific Ocean than from the Atlantic Ocean?

L115: “the gross overall nitrate/nutrient supply to the Southern Ocean surface was reduced.” Could you clarify what this sentence means?

Figure4: Modify From PO4–3 to PO43–

Conclusion:

Major revisions required.

This study presents a novel perspective on the role of NPIW in glacial CO₂ drawdown, but key aspects require clarification. Specifically, the study should explicitly define the analyzed Southern Ocean region, improve comparisons with previous modeling studies, and provide Δ¹⁴C or ideal age tracers to assess ventilation changes. Additionally, quantifying global surface pCO₂ changes and clarifying whether NPIW alone can explain the Southern Ocean carbon reservoir would strengthen the conclusions. Improving figure readability would also enhance clarity.

Reference:

Chikamoto, M. O. et al. Variability in North Pacific intermediate and deep water ventilation during Heinrich events in two coupled climate models. *Deep Sea Research Part II: Topical Studies in Oceanography* vols. 61–64 114–126 (2012).

Menviel, L., Timmermann, A., Mouchet, A. & Timm, O. Meridional reorganizations of marine and terrestrial productivity during Heinrich events. *Paleoceanography* vol. 23 (2008).

Sigman, D. M. et al. The Southern Ocean during the ice ages: A review of the Antarctic surface isolation hypothesis, with comparison to the North Pacific. *Quaternary Science Reviews* vol. 254 106732 (2021).

(Remarks on code availability)

I have access to the data and code, but there doesn’t seem to be a README or any documentation to help me run it.

Reviewer #2

(Remarks to the Author)

Summary

Shankle and co-authors proposed that the reduction of carbon and nutrients in the subsurface water in the North Pacific Ocean explains the reduced load of carbon and nutrients of the Southern Ocean surface ocean during glacial times, alongside changes in local physical processes. They provided evidences from proxy records, showing that nutrients and primary production were clearly lower during glacial periods than inter-glacial periods in both the subarctic North Pacific and the Antarctic zone of the Southern Ocean. Reasons for this reduced carbon and nutrient load were explored based on results from the Earth System Model LOVECLIM and cGENIE, and from a hosing experiment conducted with another ESM UVic. All model results confirmed that a stronger ventilation of the subsurface North Pacific leads to a reduction of carbon and nutrients loaded in the intermediate water which moves southwards and supplies the surface Southern Ocean with carbon and nutrients. This study concludes that not only local physical processes but also the biogeochemistry of the source water are important to explain the lower load of carbon and nutrients of the surface Southern Ocean and the reduced outgassing during glacial periods.

This study provides a novel aspect to explain changes in the marine carbon and nutrient cycle during glacial times. And proxy records and the load and transport of carbon and nutrients in models were analysed and demonstrated properly. The manuscript is well-structured and well-written. However, the importance of remote processes or biogeochemistry of source waters was examined very qualitatively. It would substantially increase the impact of this study when a quantitative

comparison between remote and local processes in regulating CO₂ outgassing in the Southern Ocean can be provided. Furthermore, I am not fully convinced by the arguments how the changed load of carbon and nutrients affects the efficiency of biological carbon pump in different regions (see the general comments) and more details for some parts of 'Methods' would make the manuscript easier understandable for the readers. Therefore, I can't support publication at this stage but like to encourage the authors to address the issues listed below in the revised manuscript.

General comments

1. Concept of the efficiency of biological carbon pump

It is highlighted in the manuscript that a reduced load of carbon and nutrients to the surface Southern Ocean improves the efficiency of biological carbon pump (e.g. L57-61, L394-396).

To my knowledge, the efficiency of marine biological carbon pump is a measure how effectively biological processes transport carbon from the surface to the deep ocean and often determined by the fraction of export production in primary production, or the fraction of export production that reaches the deep ocean (>1000m). One can also quantify the efficiency through the fraction of new production (primary production supported by preformed nutrients, not recycled nutrients) in total primary production. A clear definition of the efficiency of marine biological carbon pump would help to understand the connection between carbon and nutrient load and efficiency of biological carbon pump. I would like to see a quantitative analysis supporting an improved efficiency as well. Without these, it is not clear why a lower carbon/nutrient load necessarily improves the efficiency of biological carbon pump.

2. Regions of high and low efficiency of biological carbon pump

The primary production in the Southern Ocean today is not proportional to the supply of macronutrients to the SO surface ocean. Thus, those macronutrients are not efficiently consumed there. However, due to the high nutrient supply and the high fraction of diatoms in primary producers there, it is typically recognised as a region with high transfer efficiency (according to the definition how much carbon is transported from the surface to the deep ocean through biological processes). On the other hand, upwelling regions at low latitudes are due to higher temperature (and thus higher remineralisation) and higher fraction of smaller phytoplankton typically recognised as regions with lower efficiency of biological carbon pump. Furthermore, with respect to CO₂ outgassing, the Southern Ocean acts as a net CO₂ sink although outgassing takes place in some regions with strong upwelling; while low latitudes such as tropics and upwelling regions are rather CO₂ sources to the atmosphere and the subtropical gyres rather CO₂ sinks, however with extremely low biological productivity. Almost the opposite is argued in the discussion (L.336-338) which is not true generally for the Southern Ocean and the low latitudes. Please differentiate more carefully the relevant regions here and clarify the criterion for a high efficiency. And without any quantitative analysis (e.g., how much is transported to low latitudes and how it affects the export production there), I find it too speculative to say that the shift of carbon and nutrients from the Southern Ocean upwelling region to the low latitudes can improve the efficiency of biological carbon pump and thus lower the atmospheric CO₂.

3. Figures

The manuscript and SI only contain anomalies found in model results. For quantities which can be positive or negative, e.g. air-sea flux, it would be good to show the absolute values as well.

The enhanced NPIW formation is the basic for the entire analysis of carbon and nutrient load and thus should be shown, at least in SI. And please consider adding a comparison of NP ventilation and AMOC between different models to explain why different models are used and why only focusing on experiments with one model...

4. Methods and model description

It is a bit confusing to find results of three different models (Fig.3) but only the model description of one model in 'Methods'. And please explain why different models are needed for this study and what the major differences between them (see above).

It is not clear if the perturbation experiments with different models (e.g. Fig. 3) were all done in the exactly same way, since only UVic-ctrl and UVic-NP are described in the manuscript.

A brief description which components and processes of the marine carbon cycle are included in UVicESM would help to understand the analysis of model results. For example, changes in net primary production were discussed. But how it is calculated in the model? Is it only controlled by PO₄ as the only nutrient or are other nutrients involved as well? Which forms of carbon are considered in the model? I am not asking for a detailed description of the whole ESM but only for components and processes which are important for this study.

'LGM-like boundary conditions' for the experiments need to be specified. And, is the atmospheric CO₂ interactively coupled (also for radiative forcing)? L206 mentioned 'more LGM-like simulations' from Rae et al. (2020). Please explain more in detail the differences between these and the simulations conducted in this study and why the simulations in this study were

not conducted under the 'more LGM-like conditions'.

Where and how large is the area receiving freshwater hosing?

Are model results of the last model year (without inter-annual variability) of both runs compared or the averages over some of the last years?

Please clarify if the calculation of PCO₂ was done in the same way for all models.

Minor comments

L152: Should be 'LOVECLIM?'

L246: If I understood correctly, UVic-ctrl was run under LGM conditions. Why is it compared with PI PCO₂? And, does Fig.1 show PI or present-day values?

L276: 'varies by only...': increase or decrease? +1.5%?

L281: It would be helpful to see where biological productivity changes in a figure and how it compares to changes in PO₄.

L353-355: Here it is not clear in which vertical layer the enhanced (or not) stratification and ventilation is meant. Enhanced stratification between intermediate waters and deep waters does not necessarily contradict enhanced ventilation of subsurface waters.

L360: That models do not capture a feature (e.g. enhanced stratification) is not a strong argument that it unlikely happened. Furthermore, would it be better to involve more recent results from PMIP4?

I recommend at this stage to focus on the major issues given in the general comments and would be happy to provide more detailed comments after the revision.

(Remarks on code availability)

The model code is provided but the scripts to analyse model results. Since the model results will be provided by the corresponding author upon reasonable request, I was not able to run the model or test the scripts. If the model results will be provided in the revised manuscript, I will review the scripts for analysis.

Version 1:

Reviewer comments:

Reviewer #1

(Remarks to the Author)

General Evaluation:

The authors have provided thoughtful and detailed responses to each question. The revised figure captions are clearer and more informative. The concept of potential CO₂ is particularly interesting, as it offers a novel perspective on the capacity of different water masses to absorb atmospheric CO₂.

One important point concerns the interpretation of the timing of the numerical experiments in relation to paleoceanographic records. The authors analyze sediment data and highlight biogeochemical changes in the North Pacific and Southern Ocean during the Last Glacial Maximum (LGM). Meanwhile, the freshwater perturbation experiments in this study appear to be motivated by the intent to investigate the potential strengthening of NPIW during the LGM, as mentioned in the paragraph around Line 247.

If this is the case, does the study imply that many existing LGM control simulations fail to capture both the weakening of the AMOC and the strengthening of NPIW? This could potentially be an important message of the paper and might offer valuable insight into the current limitations of LGM modeling efforts.

While the manuscript is generally in good shape, a few points—particularly regarding clarity of model descriptions, consistency of units, and interpretation of key results—still require revision before the manuscript can be considered ready for publication.

Specific Comments:

Line 184:

“the long-term equilibrated response to ventilation of the North Pacific”

Are the spin-up durations consistent across all models? For example, Supplementary Figure 4 shows that cGENIE was spun up for ~1000 years — clarification would be helpful.

Line 227:

Only LOVECLIM v1.1 is explicitly mentioned here, but the study also analyzes UVic ESCM v2.9. The overturning value of ~14.8 Sv corresponds to the UVic simulation, not LOVECLIM. Please clarify.

Line 302:

"Our results therefore perhaps represent a conservative estimate of the effect of North Pacific ventilation on North Pacific and Southern Ocean biogeochemistry."

That said, the degree of NPIW strengthening varies across models — UVic appears to exhibit the strongest response. For example, Supplementary Figure 6 shows a ~3000-year difference in deep-minus-surface age, which raises the question of whether such changes are also observable in foraminiferal records.

Supplementary Figure 5:

Please clarify the mechanism behind the positive air-sea CO₂ flux anomaly in the North Atlantic (UVic-NP minus UVic-ctrl). What physical or biogeochemical processes are responsible?

Supplementary Figure 8:

Caption:

"Phosphate, (mol kg⁻¹)" → Should be "μmol kg⁻¹"?

"NPP, (mol N m⁻³ yr⁻¹)" → Should this be "mol N m⁻² yr⁻¹"?

Please verify the units.

(Remarks on code availability)

The necessary data for visualization, along with the corresponding analysis and plotting script, have been carefully reviewed, and the README provides clear instructions for usage.

Reviewer #2

(Remarks to the Author)

I much appreciate that the authors addressed all my comments and provided a lot more analyses, figures and model descriptions. However, based on the new material three major issues have been raised during the review.

1) The central finding of this study is that a stronger NP ventilation during glacials lowers the carbon and nutrient supply to the surface Southern Ocean through NPIW, and thus contributes to glacial CO₂ drawdown. So far this study shows lower concentrations of carbon and nutrients in NPIW in simulations with well-ventilated NP, and also in intermediate waters in the South Pacific, and lower concentrations of carbon and nutrients in the SO surface waters. However, the main source of carbon and nutrients to the SO surface waters is the upwelling of Circumpolar Deep Water (CDW). NPIW flows southward and reaches the tropical surface by upwelling. It is not clear to me how NPIW can reach the SO surface directly. I believe that a stronger PMOC and thus transport of carbon and nutrient-poor surface waters to the deep ocean could contribute to a lower carbon and nutrient load in CDW and then affect the outgassing in the SO. But this should happen rather through the mixing of NPIW with deeper water masses and the deep circulation pathway and upwelling in the SO. Thus, the linkage between reduced carbon and nutrients in NPIW and SO surface waters need to be thoroughly discussed and more analysis of deep mixing and circulation which in my opinion might transport the signal in carbon and nutrients from NPIW to SO surface is necessary. Additionally, the upwelling of CDW brings signals from other ocean basins as well, for example, the signal from the Atlantic which is strongly affected by AMOC weakening in this study. How large the contribution of NP ventilation is, needs to be better quantified.

If in simulations the SO surface water is supplied with carbon and nutrients in a different pathway than through CDW, this needs to be shown, explained and confirmed by proxy data.

2) In SI, Figure 2 and Table 1 show that the perturbed simulation with UVic is quite far from a LGM state and rather represents the onset of deglaciation, such as Heinrich Stadial 1. I am wondering how the results from this simulation can help to understand mechanisms driving the glacial CO₂ drawdown. The authors addressed this in the revised manuscript that they like to focus on the impact of ventilated NP on the SO biogeochemistry. However, the ocean is connected which is particularly true for the Southern Ocean where waters from all ocean basins are joined and mixed, as the authors also pointed out in the introduction. If the focus of the study is the contribution of NP ventilation to glacial CO₂ drawdown, a simulation like LOVECLIM-LGM with a stronger PMOC and a weaker AMOC but not an AMOC shutdown would be more plausible. The anomaly of carbon and nutrients in the SO surface/upwelling waters (Fig.3 and SI Fig.2) however, seems to be negligibly small, not supporting the hypothesis that a well-ventilated GLACIAL NP could account for the reduction of carbon and nutrients in the SO surface.

3) Figure 9 in SI show the air-sea CO₂ fluxes. I guess that the correct unit is molC/yr as in the caption, not as at the Y-axis. The global flux (the sum of the area below or above the curve) seems to be a net outgassing. It is even clearly the case in the control simulation. Is it still in a transient state? This is also reflected in SI Figure 5 where the absolute fluxes are shown. Although it is stated in Line 518-520: 'CO₂ was set to be prognostic in these simulations, though the LGM-like background state was first established by forcing the model under LGM CO₂ (192 ppm) until a quasi-equilibrium was reached'. Furthermore, a CO₂ decrease of 6 microatm is reported in the study from the control to perturbation experiment. How can it be explained by the strong outgassing trend from the ocean? If the air-sea and air-land gas exchange are both considered and coupled in the ESM simulations, could changes in land carbon explain this CO₂ decrease? For these questions, the temporal evolution of pCO₂ in the atmosphere during the entire simulation would help to check if the land-ocean-atmosphere gas exchange approaches an equilibrium, and the same for the total land carbon and ocean carbon pool.

Since these issues are more related to the basic concept of this study, I would encourage the authors to refine their concept and provide stronger evidence for their hypothesis. More detailed comments will be provided after these major issues have

been addressed.

(Remarks on code availability)

Version 2:

Reviewer comments:

Reviewer #1

(Remarks to the Author)

The noteworthy result of this study is that it demonstrates how the strengthening of deep circulation in the North Pacific reduces the supply of nutrients and carbon to the Southern Ocean through remote effects, thereby suppressing CO₂ outgassing from the Southern Ocean. Importantly, the study offers a new perspective on past North Pacific circulation, supported by proxy evidence.

This work is significant to the field as it addresses a gap in our understanding of the role of North Pacific deep circulation in regulating Southern Ocean CO₂ outgassing.

The results and figures presented are clear and, following revision, effectively support the authors' conclusions and claims. There are no flaws in the data analysis, interpretation, or conclusions that would prohibit publication. The methodology is sound, meets the expected standards of the field, and is described in sufficient detail to allow for reproducibility.

It is my understanding that clarifying the specific periods during which this mechanism operated in the past, as well as the timescales involved, remains an important subject for future investigation.

(Remarks on code availability)

Reviewer #2

(Remarks to the Author)

The authors provided thorough and insightful responses to my comments, clarifying key terminology, correcting and supplemented figures with deeper analyses, and clearly outlined the goals and limitations of the study. I fully agree that the perturbation experiments presented here make a valuable contribution to understanding how stronger North Pacific ventilation influences carbon and nutrient supply to the surface Southern Ocean.

While I remain not fully convinced that these simulations directly help to explain glacial CO₂ drawdown—given that they may not fully represent a glacial ocean state, the study nevertheless offers important novel perspectives and useful directions for future research.

For these reasons, I support the publication in Nature Communications.

Minor comments:

L23: '...global nutrients, carbon cycling' to 'global nutrient and carbon cycling'

L35: 'glacial-interglacial CO₂ change' to 'glacial-interglacial CO₂ variability'

L170: '...expanded NPIW cell has expanded and ...': delete the second 'expanded and'?

L180: 'Rae et al. (2020) find' to 'Rae et al. (2020) found'

L254 and L258: are the parentheses before 'After the 1000-year' and after 'a fully equilibrated state' necessary?

(Remarks on code availability)

REVIEWER COMMENTS AND COAUTHORS' RESPONSES FOR:

“Southern Ocean CO₂ outgassing and nutrient load reduced by a well-ventilated glacial North Pacific”

Madison G. Shankle; Graeme A. MacGilchrist; William R. Gray; Casimir de Lavergne; Laurie C. Menviel; Andrea Burke; James W. B. Rae

Revisions Round 1 – 2025 April 25

REVIEWER #1

General Evaluation:
Summary and Strengths of the Study: This study investigates the mechanisms behind low atmospheric CO₂ levels during glacial periods by analyzing the relationship between the Southern Ocean and the North Pacific intermediate waters using both models and proxy data. The authors propose that the expansion of North Pacific intermediate water (NPIW) during glacial periods may have led to a decrease in carbon and nutrient concentrations in the surface waters of the Southern Ocean, thereby suppressing CO₂ release. This perspective extends beyond the commonly discussed mechanisms of Southern Ocean stratification and westerly wind shifts, introducing a remote influence via intermediate waters. A key strength of the study is the use of multiple models to examine the response of NPIW to freshwater input in the North Atlantic. This approach allows for a robust assessment of how North Pacific intermediate circulation could contribute to global carbon cycle changes. Additionally, the study is well-situated within the existing literature, particularly in its focus on how changes in the intrinsic carbon and nutrient concentrations of water masses upwelling in the Southern Ocean may have played a crucial role. The motivation and research objectives are clearly defined. We thank Reviewer 1 for their insightful and constructive review. Their comments below have strengthened our narrative and analysis, and we have worked carefully to address them. In particular, we have given special attention to the concerns on clarity (e.g., explicitly referring to specific Southern Ocean regions, demonstrating ventilation with a new figure of radiocarbon age, and placing our results in better context with existing literature) and on low-latitude and North Atlantic surface PCO₂ and flux changes and what these mean for our interpretations. Please find our point-by-point responses below.

Areas for Improvement:
Conceptual Clarity and Positioning: R1 Comment 1/14: The study does not clearly define the specific region of the Southern Ocean being examined. Based on the text, it appears that the focus is on the subantarctic zone (approximately 45°S–55°S). Given that previous proxy-based studies (e.g., Sigman et al., 2021) have discussed regional differences in biological production and nutrient consumption efficiency between the subantarctic and Antarctic zones, the study should clarify which region its conclusions apply to. If the analysis aims to track changes along neutral density surfaces of approximately 27.5–28 kg m⁻³ in the modern ocean, this should be explicitly stated. • Following the reviewer's suggestion, we have added detail to our discussion of Southern Ocean anomalies throughout the text to clarify their respective regions of influence. Specifically that while anomalies in tracers such as nutrient concentrations and PCO₂ span both the SAZ and AZ (spanning from ~40°S to the Antarctic continent),

outgassing anomalies are strongest at the core of the upwelling North Pacific isopycnals (e.g., $\sigma_2 = 36.5 \text{ kg m}^{-3}$ in Fig. 5) and in the SAZ (~40-60°S), with weakened anomalies to the south due to sea ice cover.

- In-text edits can be found in the Abstract (lines ~33-34), the results (first and third paragraphs of “Low PCO_2 drives reduced Southern Ocean outgassing”, commencing on lines ~274 and ~293), and the Conclusion (line ~462)

R1 Comment 2/14: The response of North Pacific intermediate ventilation to North Atlantic freshwater forcing appears to differ between models. Some models show little strengthening of NPIW in response to freshwater input into the North Atlantic (e.g., Chikamoto et al., 2012). The study finds that MIROC simulations exhibit a decrease in nutrient concentrations from NPIW to the Southern Ocean, attributed to a reduction in the nutrient content of waters originating from the south. This interpretation differs from this study. To better support the conclusions regarding the role of NPIW in altering nutrient supply, a more thorough comparison with past studies is necessary. The current citations and discussions on this aspect are insufficient.

- The reviewer is correct in noting there is inter-model differences in the character of North Pacific ventilation in response to North Atlantic freshwater forcing, both between the models used in this paper and in those of previous studies. We now comment on these diverse responses across models at lines ~225-236. While we do not attempt to explain the cause of this inter-model spread, it being beyond the scope of this study, we nevertheless emphasize that some degree of North Pacific ventilation is a robust response across models, which is the salient feature of interest to our argument. This is true even in the Chikamoto study the reviewer highlights, in which even the MIROC simulation still sees “an enhancement of the deep sinking branch of the PMOC” and an overturning of ~5 Sv, supporting North Pacific ventilation as a robust response to North Atlantic freshwater changes. Furthermore, despite differing circulation changes, both LOVECLIM and MIROC see strong reductions in mid-depth carbon/nutrients (see their Fig. 9, below). This demonstrates that North Pacific ventilation (even if weak) is a highly effective means of reducing mid-depth carbon/nutrients, directly supporting the main motivation behind our study. Text supporting this argument has also been added at lines ~160-164.

Chikamoto et al. (2012). Fig. 9. Their caption: Nitrate concentration changes in the euphotic zone of (A) MIROC and (B) LOVECLIM (shaded, mmol m^{-3}) and the sea ice thickness in February (black contours) in the LGM hosing experiments (0.15 and 0.3 m). Zonal mean nitrate concentration changes in the Pacific Ocean of (C) MIROC and (D) LOVECLIM (shaded, mmol m^{-3}). Grey contours in Fig. 9C and D are the nitrate concentration changes at 2.0 mmol m^{-3} intervals.

- We have also added text (lines ~164-168) acknowledging that other mechanisms have been put forwards as a means of altering sub-surface nutrients (e.g., origins in the south, as in Chikamoto et al.). However it is clear in our simulations that the anomaly arises from the north with the onset of northern ventilation (see our Supplementary Figure 4) and we have chosen to focus on this mechanism given it being (a) a robust feature across models (e.g., Fig. 3) and (b) well-supported by glacial proxy data (e.g., Fig. 2f and references within the main text).

R1 Comment 3/14: To strengthen the interpretation of ocean circulation timescale changes, it would be helpful to present $\Delta^{14}\text{C}$ or ideal age tracers as indicators of ventilation time. This would enable comparisons with sediment records and provide a useful benchmark for evaluating model responses.

- We appreciate this suggestion and now provide a new supplementary figure of deep-minus-surface radiocarbon age and its anomaly between the UVic-NP and UVic-ctrl

simulations to better support and illustrate the ventilation changes. We refer the reader to this new supporting figure (Supplementary Figure 6, below) when describing the ventilated case of UVic, at line ~249.

- Please note that the ventilation in UVic-NP is also now supported with a new Supplementary Figure 2 of the overturning stream function, requested by Reviewer 2.
- The presentation of radiocarbon age we provide is equivalent to benthic-planktic foraminifera radiocarbon ages. However, it is not possible to make a direct comparison to glacial proxy data since in these simulations ^{14}C in the atmosphere was allowed to vary rather than being kept constant at glacial values. Furthermore, these simulations were not run with the goal of reproduction glacial ^{14}C distributions, which will depend on many factors (e.g., Southern Ocean circulation, sea ice changes, etc.), but instead are used to investigate the impact of North Pacific ventilation on subsurface nutrient and carbon delivery to the Southern Ocean.

(new) Supplementary Figure 6. North Pacific ventilation is evidenced by strong reductions in deep-minus-surface radiocarbon age. Deep-minus-surface radiocarbon years (equivalent to benthic-planktic foraminifera radiocarbon ages) in (a) UVic-ctrl and (b) UVic-NP, with each grid cell's radiocarbon years value having subtracted from it the radiocarbon years value of the surface cell at its same latitude and longitude. (c) Radiocarbon years anomaly (UVic-NP – UVic-ctrl), with negative anomalies indicating younger (less radiocarbon-depleted) better-ventilated waters. Output in each panel represents averages over the last 10 years of simulation. Note, while deep-minus-surface radiocarbon years is equivalent to benthic-plankton foraminifera radiocarbon ages, it is not possible to directly compare this model data to glacial radiocarbon proxy data because of how atmospheric ^{14}C was allowed to vary in the model.

Methodology:

R1 Comment 4/14: In the paragraph starting at L290 and in Supplementary Figure 4, the study attempts to disentangle the factors contributing to the pCO_2 anomaly. However, the

method used to separate the contributions of sea ice and wind speed is not sufficiently explained. Providing more details on this methodology would enhance clarity.

We now include more description in the main text describing how this decomposition of the flux anomaly is achieved. In short, flux is first computed using the field for a variable of interest (e.g., wind speed) as simulated by UVic-NP and using fields for all other relevant variables from UVic-ctrl. The resulting flux field then has subtracted from it the flux field obtained from UVic-ctrl (computed using fields for all variables from UVic-ctrl). The resulting anomaly thus gives the flux anomaly due only to changes in the variable of interest from UVic-ctrl to UVic-NP.

Text describing this methodology has been added at the paragraph flagged by the reviewer (at lines ~344-351 in the main text) as well as in the caption for this figure (note that it has been re-numbered to Supplementary Figure 7, see lines ~95-100 in the Supporting Information).

Results and Interpretation:

R1 Comment 5/14: Supplementary Figure 5 shows that when NPIW strengthens, the immediate response includes an increase in surface nutrient and carbon concentrations. Would this increase in nutrients not contribute to higher $p\text{CO}_2$ in low-latitude surface waters, potentially offsetting the negative $p\text{CO}_2$ anomaly in the Southern Ocean? This point needs to be examined further to ensure that the proposed mechanism effectively explains the observed CO_2 drawdown.

- We are grateful to the reviewer for this comment and the next, which help clarify our explanation of changes surface $p\text{CO}_2$ (note, PCO_2 by its calculation is equivalent to surface $p\text{CO}_2$).
- We have updated these figures to show, instead of DIC, the evolution of PCO_2 at depth over time (new Supplementary Figure 4), and its final state across the surface ocean (new Supplementary Figure 5). These show that surface PCO_2 (the most relevant value for atmospheric CO_2 change) decreases across virtually all of the Indo-Pacific basin. While a positive anomaly in surface nutrients and DIC can be found at the onset of enhanced overturning in the North Pacific (as also discussed in Rae et al., 2014, 2020), this effect is transient and short lived, disappearing once nutrient-rich waters initially in the subsurface are replaced by better ventilated waters. We address the implications of these PCO_2 results for global fluxes and atmospheric CO_2 change with edits discussed in the comment below.

(New) Supplementary Figure 4. The time-evolution of potential $p\text{CO}_2$ (PCO_2) (μatm) in UVic-NP shows carbon accumulating in deep waters and negative anomalies developing from the north. (a-e) Absolute PCO_2 (see Methods) (μatm) along 160°W over the 1000 years of the UVic-NP simulation. (f-j) PCO_2 anomaly relative to UVic-ctrl (μatm) over the same time.

(New) Supplementary Figure 5. Negative anomalies in potential $p\text{CO}_2$ (PCO_2) pervade the Indian-Pacific basin and Southern Ocean, driving negative anomalies in air-sea CO_2 flux in key outgassing regions. PCO_2 (see Methods) (μatm) (a, b) and air-sea CO_2 flux ($\text{mol m}^{-2} \text{yr}^{-1}$) (d, e) in UVic-ctrl (a, d), UVic-NP (b, e), and as anomalies (UVic-NP – UVic-ctrl) (c, f). Anomalies compare averages over the last 10 years of each simulation. Negative flux anomalies indicate reduced CO_2 outgassing or enhanced CO_2

uptake. Regions of positive outgassing anomalies in the Indian-Pacific basin correspond with regions of weakly-negative or zero PCO_2 anomaly, indicating they are driven by the slightly decreased atmospheric pCO_2 in UVic-NP ($\sim 185 \mu\text{atm}$) relative to UVic-ctrl ($\sim 191 \mu\text{atm}$). This likely also contributes to the positive flux anomalies in the North Atlantic, though these are expected to be primarily a result of the freshwater forcing applied in this region and resulting reduction in overturning, rather than a direct effect of the North Pacific ventilation itself.

R1 Comment 6/14: Although the study demonstrates that Southern Ocean pCO_2 decreases, low-latitude and North Atlantic surface pCO_2 increases (Supplementary Figure 3). This raises the question of whether a strengthened NPIW transported CO_2 -rich waters from the North Pacific and subantarctic regions to other ocean basins via surface pathways. To strengthen this argument, the study should quantify global-scale surface ocean pCO_2 changes.

- We hope the following explanation and revisions clarifies the signals we see in our output and how we interpret them in the context of global-ocean and atmospheric change.

Carbon-rich waters to N.Atl and low-latitudes via surface pathways

- The Supplementary Figure 3 (from our initial submission) that the reviewer references depicts flux (outgassing) anomalies, not PCO_2 . While fluxes do see such positive anomalies, PCO_2 is reduced across large swathes of the surface Indo-Pacific, as illustrated in the new Supplementary Figure 5 above (panel c). (The Atlantic will be discussed below.)
- The positive flux anomalies in low latitudes the reviewer was originally pointing to are not a result of a redistribution of oceanic carbon due to circulation change, but rather the slight decrease ($\sim 6 \mu\text{atm}$) in atmospheric CO_2 between the control ($\sim 191 \mu\text{atm}$) and ventilated ($\sim 185 \mu\text{atm}$) simulation. With surface water PCO_2 changed only slightly or not at all in these regions, and with atmospheric CO_2 overall lower, these regions display positive outgassing anomalies.
- As for the positive flux anomalies in the North Atlantic, these are most likely to be a direct result of the reduced AMOC in this experiment (now quantified in Supplementary Table 1 and Supplementary Figure 2), rather than an effect of the remote N. Pacific ventilation. The freshwater forcing reduces NADW formation, resulting in a sluggish AMOC and poorly-ventilated carbon-rich subsurface waters which supply the outgassing – akin to the modern North Pacific.
- To help clarify these points, we have replaced the old Supplementary Figure 3 (only depicting flux anomalies) with the new Supplementary Figure 5 above (presenting PCO_2 and flux anomalies), providing a clearer presentation of and distinction between surface PCO_2 and flux anomalies for the reader. We also in the caption of this new figure describe the North Atlantic changes.

Global-scale surface ocean PCO_2 and flux changes

- The next question that naturally arises from our results is whether PCO_2 or flux anomalies cancel out in the global integral, as this will be the quantity relevant to atmospheric CO_2 change. In short, they do not cancel out and in the net are in the right direction for drawing down atmospheric CO_2 .
- To illustrate this, we now present Indo-Pacific-integrated fluxes alongside the globally-integrated fluxes the reviewer requested (new Supplementary Figure 9, below). In the Indo-Pacific (panel a), it can be seen that regions of net outgassing in the control (blue line) (such as the equatorial latitudes and the Southern Ocean) see reduced outgassing in the ventilated simulation (orange line). The high northern latitudes of the Pacific also become stronger in their ingassing (more negative values of flux). In total (integrating across latitude and longitude), UVic-NP sees a $\sim 29.6\%$ decrease in total flux (mol yr^{-1}) across the Indian-Pacific basin relative to UVic-ctrl.
- The same pattern is also exhibited in the global integral (panel b), even when including the positive flux anomalies of the North Atlantic (new Supplementary Figure 5f) – which we don't think are representative of an LGM-like state (where there is upper ocean ventilation in both the North Atlantic and North Pacific). In the global integral, UVic-NP sees an $\sim 11.6\%$ decrease in total outgassing relative to UVic-ctrl. We believe this net reduction in outgassing drives the $\sim 6 \mu\text{atm}$ drawdown observed between the UVic-ctrl and UVic-NP simulations.

- In the main text, we avoid placing too much emphasis on this quantified global flux change, instead focusing on the basin-specific influence of circulation changes on biogeochemistry in the Indo-Pacific. This reason for this is that, by nature of the applied forcing (which leads to simultaneous and convoluting large-scale changes in the North Atlantic), the simulations presented here are not ideally configured to provide direct insight on the global response to N. Pacific ventilation alone (given the concurrent reduction of overturning in the Atlantic in these simulations). Hence, we wish to avoid confusion for the reader and not suggest that we have unambiguously quantified the global effect of this N. Pacific ventilation mechanism.
- A model experiment designed to isolate and quantify the effects of N. Pacific ventilation alone on ocean and atmospheric carbon should induce such ventilation changes without concurrent changes in the Atlantic. However, such forcing is challenging to implement in coupled Earth system models in which various components of the Earth system interact in complex non-linear ways. Such work, while a valuable avenue for future investigation, is beyond the scope of our present study. Future studies may seek to isolate and quantify the atmospheric-drawdown effect of North Pacific ventilation more directly.
- Nevertheless, we thank the reviewer for encouraging us to address this question, as it certainly better illustrates the broader implications of our results for global change. This new quantification and discussion better supports our conclusion of North Pacific ventilation having implications for glacial CO₂ drawdown. We have now included text discussing this and referring the reader (lines ~406-413) to these new Supplementary Figures 5 and 9 and information in their captions.

(new) Supplementary Figure 9. North Pacific ventilation reduces outgassing in high latitude (>40°N/S) and equatorial (±20°) regions. Zonally-integrated upwards CO₂ flux (mol yr⁻¹, with positive values denoting flux out of the ocean) as a function of latitude, integrated across the (a) Indian-Pacific basin and (b) entire globe. Flux in UVic-ctrl and UVic-NP are shown in blue and orange, respectively, with the anomaly (UVic-NP – UVic-ctrl) plotted in the dashed black line. Negative values of the anomaly (grey shading) indicate either reduced outgassing or enhanced ingassing. In total (integrated zonally and meridionally, excluding the Arctic Ocean >65°N), UVic-NP exhibits a ~29.6% decrease in total flux (mol yr⁻¹) relative to UVic-ctrl across the Indo-Pacific basin and an ~11.6% decrease globally (the smaller value being due to the concurrent effects of reduced overturning in the Atlantic (a result of the freshwater forcing, see Methods) inducing some positive flux anomalies there, see Supplementary Fig. 5). More confidence is therefore placed in the quantification of Indian-Pacific anomalies than global anomalies, given the convoluting effect of the reduced Atlantic overturning impacting fluxes in that basin. A zonal integral across the Indian-Pacific basin was computed by masking the data to only this basin using CMIP6 basin codes available through the “cmip_basins” Python package (https://github.com/jkrasting/cmip_basins).

R1 Comment 7/14: The study contributes to the “Polar Twins” framework, but it remains unclear whether the authors consider changes in NPIW alone to be sufficient in explaining the large carbon reservoir in the Southern Ocean. Alternatively, are additional processes, such as deep circulation changes and stratification, still necessary? Southern Ocean deep waters are characterized by high salinity, low δ¹³C, and depleted Δ¹⁴C in proxy records—can these be explained without invoking stratification? Addressing this point would clarify the broader implications of the study’s findings.

- We thank the reviewer for this point, which will better situate this North Pacific mechanism into a more comprehensive view of Southern Ocean changes.
- We have added text to clarify that we see this as a contributing mechanism to glacial ocean carbon storage. While NPIW ventilation impacts ocean-atmosphere carbon partitioning (modulating the carbon content of important surface waters interfacing with the atmosphere), additional processes likely contributed to retaining this carbon in the ocean.

These processes, supported by proxy data such as those cited by the reviewer, include changes in sea ice and disequilibrium carbon (Stephens & Keeling, 2000; Marzocchi and Jansen, 2019; Khatiwala et al. 2019), shifts in bottom water properties and stratification (Adkins et al., 2002; Ferrari et al., 2014; Jansen & Nadeau, 2016; Nadeau et al., 2019), as well as iron fertilization (Bopp et al., 2003; Kohfeld et al., 2005; Lambert et al., 2015; Thöle et al., 2019). Highlighting the role of NPIW ventilation alongside other contributing mechanisms to glacial change fits naturally into our discussion, and the new text can be found at lines ~417-421. Beyond this, determining the magnitude of glacial change from NPIW ventilation versus other factors remains a challenge requiring dedicated future work which we hope our results motivate.

Presentation and Writing

R1 Comment 8/14: The figure readability should be improved. Axis labels and contour labels should be enlarged for better visibility.

All figures have been revisited to improve readability, with special attention given to axis and contour label font sizes.

R1 Comment 9/14: Figure 2: Additional explanations should be provided on what the proxy data represent (e.g., whether higher/lower values indicate increased or decreased biological productivity or nutrient depletion).

This explanation has been added to Figure 2's caption (lines ~570-571).

R1 Comment 10/14: Figures 3 and 4: Contour line colors and label sizes should be adjusted to enhance visibility.

Contour line colors and label sizes have been improved in all relevant figures (Figures 3, 4, and 5, and Supplementary Figures 3 and 7).

Specific comments

R1 Comment 11/14: L67: "Today the Southern Ocean receives most of its carbon and nutrients from mid- depths (1-3 km) of the Pacific basin⁹, where poor ventilation allows for their accumulation¹⁵." Which ocean region does the Southern Ocean refer to? Also, does "Most" imply that nutrient supply comes more from the Pacific Ocean than from the Atlantic Ocean?

We have clarified that we mean Southern Ocean surface waters (both SAZ and AZ) and that more carbon/nutrients come from the Pacific than Atlantic with edits made to lines ~69-72.

R1 Comment 12/14: L115: "the gross overall nitrate/nutrient supply to the Southern Ocean surface was reduced." Could you clarify what this sentence means?

We have rephrased this sentence to be clearer, stating that greater nutrient consumption during glacial times can only be reconciled with reduced productivity if productivity had a smaller overall pool or inventory of nitrate available to it. This signals that the nutrient supply to the Southern Ocean surface was reduced during glacial periods. Edits to the text have been made at lines ~114-117.

R1 Comment 13/14: Figure4: Modify From PO4-3 to PO43-

This has been amended (line ~609).

Conclusion

Major revisions required.

This study presents a novel perspective on the role of NPIW in glacial CO₂ drawdown, but key aspects require clarification. Specifically, the study should explicitly define the analyzed Southern Ocean region, improve comparisons with previous modeling studies, and provide $\Delta^{14}\text{C}$ or ideal age tracers to assess ventilation changes. Additionally, quantifying global surface pCO₂ changes and clarifying whether NPIW alone can explain the Southern Ocean carbon reservoir would strengthen the conclusions. Improving figure readability would also enhance clarity.

Reference

Chikamoto, M. O. et al. Variability in North Pacific intermediate and deep water ventilation during Heinrich events in two coupled climate models. *Deep Sea Research Part II: Topical Studies in Oceanography* vols. 61–64 114–126 (2012).

Menviel, L., Timmermann, A., Mouchet, A. & Timm, O. Meridional reorganizations of marine and terrestrial productivity during Heinrich events. *Paleoceanography* vol. 23 (2008).

Sigman, D. M. et al. The Southern Ocean during the ice ages: A review of the Antarctic surface isolation hypothesis, with comparison to the North Pacific. *Quaternary Science Reviews* vol. 254 106732 (2021).

Thank you again for your thoughtful and constructive critique of our manuscript. We are grateful for your comments, which we believe have improved the clarity and rigor of the study.

Reviewer #1 (Remarks on code availability)

R1 Comment 14/14: I have access to the data and code, but there doesn't seem to be a README or any documentation to help me run it.

We have now written a ReadMe document describing the analysis scripts supporting this study and how to run them. It has been made available to the reviewers on the Code Ocean capsule and will be made available to readers alongside the code upon publication.

REVIEWER #2

Summary

Shankle and co-authors proposed that the reduction of carbon and nutrients in the subsurface water in the North Pacific Ocean explains the reduced load of carbon and nutrients of the Southern Ocean surface ocean during glacial times, alongside changes in local physical processes. They provided evidences from proxy records, showing that nutrients and primary production were clearly lower during glacial periods than inter-glacial periods in both the subarctic North Pacific and the Antarctic zone of the Southern Ocean. Reasons for this reduced carbon and nutrient load were explored based on results from the Earth System Model LOVECLIM and cGENIE, and from a hosing experiment conducted with another ESM UVic. All model results confirmed that a stronger ventilation of the subsurface North Pacific leads to a reduction of carbon and nutrients loaded in the intermediate water which moves southwards and supplies the surface Southern Ocean with carbon and nutrients. This study concludes that not only local physical processes but also the biogeochemistry of the source water are important to explain the lower load of carbon and nutrients of the surface Southern Ocean and the reduced outgassing during glacial periods.

This study provides a novel aspect to explain changes in the marine carbon and nutrient cycle during glacial times. And proxy records and the load and transport of carbon and nutrients in models were analysed and demonstrated properly. The manuscript is well-structured and well-written. However, the importance of remote processes or biogeochemistry of source waters was examined very qualitatively. It would substantially increase the impact of this study when a quantitative comparison between remote and local processes in regulating CO₂ outgassing in the Southern Ocean can be provided. Furthermore, I am not fully convinced by the arguments how the changed load of carbon and nutrients affects the efficiency of biological carbon pump in different regions (see the general comments) and more details for some parts of 'Methods' would make the manuscript easier understandable for the readers. Therefore, I can't support publication at this stage but like to encourage the authors to address the issues listed below in the revised manuscript.

We are grateful to Reviewer 2 for their constructive and thoughtful feedback of our manuscript. Addressing the comments below has provided clarity and more detailed support on many crucial points in our study. We are particularly grateful for the insight from Reviewer 2 on biological pump efficiency and their suggestions to enhance our discussion around this. We have worked carefully to address this and the other points raised. Please find our point-by-point responses to their comments below.

General comments

1. Concept of the efficiency of the biological carbon pump

R2 Comment 1/18: It is highlighted in the manuscript that a reduced load of carbon and nutrients to the surface Southern Ocean improves the efficiency of biological carbon pump (e.g. L57-61, L394-396).

To my knowledge, the efficiency of marine biological carbon pump is a measure how effectively biological processes transport carbon from the surface to the deep ocean and often determined by the fraction of export production in primary production, or the fraction of export production that reaches the deep ocean (>1000m). One can also quantify the efficiency through the fraction of new production (primary production supported by preformed nutrients, not recycled nutrients) in total primary production. A clear definition of the efficiency of marine biological carbon pump would help to understand the connection between carbon and nutrient load and efficiency of biological carbon pump. I would like to see a quantitative analysis supporting an improved efficiency as well. Without these, it is not clear why a lower carbon/nutrient load necessarily improves the efficiency of biological carbon pump.

- This is a very helpful comment, which highlights the various definitions of biological pump “efficiency” that exist in allied fields and the need to clarify our terminology.
- Our use of biological pump efficiency describes the extent to which biological production can “keep pace” with the carbon and nutrients supplied to it. In this sense, the Southern Ocean in its interglacial or preindustrial state is “inefficient”, in that the carbon/nutrient supply so overwhelms and outpaces biology’s ability to fix carbon that much of the carbon evades to the atmosphere (and many of the nutrients are lost to the deep ocean). By contrast, when the incoming carbon/nutrient supply is reduced (as in our simulations), the same limited levels of production can fix a greater proportion of the supplied carbon, and outgassing is reduced. In this sense, our use of “biological pump efficiency” is in line with its usage in the paleoclimate literature but we now realize could be misleading to a community concerned with the efficiency of export production.
- The changes in outgassing we describe are not a result of any changes in biological processes or rates (e.g., a changing fraction of new versus total production, or of export production in total production), but instead arise solely by nature of the altered incoming carbon/nutrient supply, which is what we set out to demonstrate that a ventilated N. Pacific would impact. N. Pacific ventilation would not be expected to alter biological processes directly or the fraction of new/total production etc. except *indirectly* through its influence on the carbon/nutrient load of supplying waters. We therefore focused our study on the direct impacts of ventilation on the biogeochemistry of waters supplied to the Southern Ocean and found this to be the major driver of the reduced outgassing signal (e.g., Supplementary Figure 7).
- For the purpose of this revision and to avoid ambiguity, we have avoided the use of the word “efficiency” and instead now discuss biology’s “ability” (or a similar word) to counteract CO₂ evasion following the upwelling of carbon (again, being careful to be clear that in our simulations this is a function of the incoming carbon supply rather than enhanced productivity rates, more efficient export production, etc.). We have been careful to re-write several parts of the text to avoid any confusion.
 - The most important edits have been made to the abstract (lines ~27-28 and ~34-35) and to the Introduction, where we introduce the importance of reducing the carbon/nutrient supply to counteracting CO₂ outgassing. This is where our use of the word “efficiency” differs from the concept of export, and we have re-written the text to instead discuss biology’s ability to fix carbon relative to the supply and counteract CO₂ evasion. Please find re-working of the text in the second paragraph of the Introduction (lines ~53-62).
 - In other places, this point was better addressed by simply replacing “biological pump efficiency” with “outgassing”, which is more accurate to what we meant to say and to what the reduced carbon/nutrient load directly impacts. For example, in the final paragraph of the Conclusion (lines ~468-479):
 - “A reduced carbon and nutrient load of the Southern Ocean surface, given its impact on biological pump efficiency, ...” has been changed to “...given its impact on outgassing, ...”. Similar clarifying edits can be found at lines ~320-322, 371-372, and ~422.

“Quantitative analysis supporting improved efficiency”

- Given the discussion above, we have not quantified processes related to export efficiency (e.g., export rates, fraction of new to total primary production, etc.), but we acknowledge the value that a more quantitative analysis of “efficiency” as we mean it would have in our manuscript. While a direct quantification is prohibitive for reasons we will discuss, we are confident that our results do support improved efficiency, in terms of increasing the fraction of carbon/nutrient supply fixed by biological production.
- While diagnostics for utilization are not available for these simulations, we can still infer that the fraction of nutrients (or equally, carbon) used by biology relative to supply is increased, given the reduced incoming supply. A reduced supply is inferred from clear decreases in nutrient/carbon concentration of upwelled waters (Fig. 5) alongside negligible changes in transport (i.e., upwelling and stratification, discussed at lines

~313-317). With productivity being roughly the same between our control and ventilated simulations (as indicated by net primary production, lines ~325-327), this reduced supply infers that the fraction of carbon/nutrients used by biology relative to the supply has been improved.

- Nevertheless, we acknowledge and appreciate the reviewer encouraging better quantitative analysis. In this light, and in addition to the quantification of outgassing reductions already in the text, we have pointed to reader to these lines of reasoning with added text at lines ~320-322 and lines ~333-335.

2. Regions of high and low efficiency of biological carbon pump

R2 Comment 2/18: The primary production in the Southern Ocean today is not proportional to the supply of macronutrients to the SO surface ocean. Thus, those macronutrients are not efficiently consumed there. However, due to the high nutrient supply and the high fraction of diatoms in primary producers there, it is typically recognised as a region with high transfer efficiency (according to the definition how much C is transported from the surface to the deep through biological processes). On the other hand, upwelling regions at low latitudes are due to higher temperature (and thus higher remineralisation) and higher fraction of smaller phytoplankton typically recognised as regions with lower efficiency of biological carbon pump. Furthermore, with respect to CO₂ outgassing, the Southern Ocean acts as a net CO₂ sink although outgassing takes place in some regions with strong upwelling; while low latitudes such as tropics and upwelling regions are rather CO₂ sources to the atmosphere and the subtropical gyres rather CO₂ sinks, however with extremely low biological productivity. Almost the opposite is argued in the discussion (L.336-338) which is not true generally for the Southern Ocean and the low latitudes. Please differentiate more carefully the relevant regions here and clarify the criterion for a high efficiency. And without any quantitative analysis (e.g., how much is transported to low latitudes and how it affects the export production there), I find it too speculative to say that the shift of carbon and nutrients from the Southern Ocean upwelling region to the low latitudes can improve the efficiency of biological carbon pump and thus lower the atmospheric CO₂.

- Again, the reviewer has illuminated a point of confusion on one of the central aspects of our study, for which we are grateful. This point again stems from our use of the term “biological pump efficiency”. We believe this point will be largely resolved by the discussion and clarity added to the text regarding the last comment, but we will also address the additional points raised here.
- The reviewer is correct that the contemporary Southern Ocean is today a net sink of CO₂, however this is a modern phenomenon arising predominantly from the anthropogenic carbon rapidly accumulating in the atmosphere today. Our focus is on glacial cycles and thus on the transition out of an interglacial (e.g., pre-industrial, un-anthropogenically-perturbed) state into a glacial state. In its unperturbed interglacial state, the Southern Ocean is believed to be a net source of “natural” carbon (Gruber et al. 2009, 2019; Menviel et al., 2023). We’ve added text to our Introduction where we introduce this concept, to avoid any confusion around this point (lines ~49-51).
Gruber, N. et al. Oceanic sources, sinks, and transport of atmospheric CO₂. *Global Biogeochem Cycles* 23, (2009).
Gruber, N., Landschützer, P. & Lovenduski, N. S. The variable Southern Ocean carbon sink. *Ann Rev Mar Sci* 11, 159–186 (2019).
Menviel, L. C., Spence, P., Kiss, A. E., Chamberlain, M. A., Hayashida, H., England, M. H., & Waugh, D. (2023). Enhanced Southern Ocean CO₂ outgassing as a result of stronger and poleward shifted southern hemispheric westerlies. *Biogeosciences*, 20(21), 4413-4431.
- Turning to the core of this comment, we agree with the reviewer’s summary of Southern Ocean versus low-latitude transfer efficiency; again the confusion here stems from a lack of clarity on the meaning of “biological pump efficiency”. Here, as described above, we are referring to the ability of biological uptake to keep up with carbon/nutrient supply, which impacts surface PCO₂ and outgassing. Where previously we distinguished between low- versus high-latitudes, we now discuss more clearly that outgassing is

reduced (due to the better-ventilation and lower carbon content of the supplying waters) in key regions of net outgassing – that is, regions where in the control state a strong carbon/nutrient subsurface supply outpaces biological carbon fixation (e.g., North Pacific and Southern Ocean, as well as the Eastern Equatorial Pacific, see Supplementary Figure 8). While some increased outgassing is seen in low-latitude regions, we make the key distinction that nutrients supplied to these regions, if not used by biology locally, are ultimately fated to stay in surface waters and drive compensatory biological carbon fixation elsewhere. By contrast, nutrients upwelled and unused in the Southern Ocean can be lost to the abyss via deep water formation and lose their compensatory potential, leading to a net increase in atmospheric CO₂ (as per Ito and Follows, 2005). This is the distinction we were making between high and low latitudes, and we hope our edits have improved the clarity of this discussion (lines ~387-403).

Ito, T., & Follows, M. J. (2005). Preformed phosphate, soft tissue pump and atmospheric CO₂. *Journal of Marine Research*, 63, 813-839.

- Regarding the reviewer's request for a more quantitative backing to our discussion, we have now done so with the quantity central to our narrative, CO₂ outgassing. This better demonstrates the implications of our results for atmospheric CO₂ with zonal and global integrals of CO₂ flux rates (Supplementary Fig. 9) (line ~120 in Supporting Information), which in the net are negative (into the ocean) in response to North Pacific ventilation. Please see our response to comments 5 and 6, under "Results and Interpretation", from Reviewer 1 for more on this figure.

3. Figures

R2 Comment 3/18: The manuscript and SI only contain anomalies found in model results. For quantities which can be positive or negative, e.g. air-sea flux, it would be good to show the absolute values as well.

We have updated our Supplementary Figure 5 (line ~66 in the Supporting Information, previously Supplementary Fig. 3) to show absolute values of air-sea flux in UVic-ctrl and UVic-NP alongside the anomaly. This version of the figure is a better illustration of and better supports our discussion of flux changes (e.g., our response to comment #2 above), so we appreciate this feedback.

R2 Comment 4/18: The enhanced NPIW formation is the basic for the entire analysis of carbon and nutrient load and thus should be shown, at least in SI. And please consider adding a comparison of NP ventilation and AMOC between different models to explain why different models are used and why only focusing on experiments with one model...

- We now include a new Supplementary Figure 2 illustrating NPIW ventilation. We also provide a new Supplementary Table 1 comparing North Pacific and North Atlantic maximum overturning stream function values between the four models depicted in Figure 3. We hope these additions will better quantify the UVic ventilation forming the basis of this study and ease model comparison for the reader. The reader is now referred to this new table and figure when discussing both the 4-way model comparison and UVic's ventilation specifically (lines ~156, 224, 242, and 250-251, as well as in Figure 3's caption at line ~595).
- We agree that a more explicit discussion of the different models in Figure 3 would be well-placed in our study. Please see our full response on this topic in the next comment below.
- There are several reasons for focusing specifically on the UVic simulations for our main analysis. First, given the range of model configurations, boundary conditions, and ventilation strengths depicted in Figure 3, we believe the reduction of mid-depth carbon/nutrients to be a robust response to ventilation, independent of model choice. It therefore would have been both unwieldy and unnecessary to show the same analysis on all four models. Such a model comparison was not the objective of our study. We

chose the UVic model for further analysis because it exhibits the clearest and most compelling signal of the enhanced ventilation, the effects of which are what we wished to explore. Furthermore, the UVic model has the highest horizontal ocean resolution of these models, and thus we expect the most faithful representation of high latitude ocean dynamics. We hope the additional text at lines ~207-216 provides clarity on our motivation to focus only on UVic, copied below for reference.

“We now quantify the impact of a well-ventilated North Pacific on Southern Ocean carbon and nutrient content in the intermediate-complexity University of Victoria Earth System Climate Model (UVic ESCM) v2.9. While mid-depth carbon/nutrient reduction is a robust response to North Pacific ventilation across the models, independent of model choice or even ventilation strength (Figure 3, see also Chikamoto et al., 2012), we chose this model to analyze in detail as it has the highest resolution (1.8° latitude x 3° longitude) compared to the other models (LOVECLIM/LOVECLIM-LGM: 3° x 3°; c-GENIE: 5° latitude x 10° longitude) and, perhaps linked to this, most clearly showcases the signal of the dynamics we set out to demonstrate, with the low-carbon, low-nutrient anomaly propagating clearly from the North Pacific and being upwelled in the Southern Ocean. We present these results as a proof of concept of a novel mechanism, one that we hope motivates future study in more targeted sets of simulations.”

Pacific Meridional Overturning Stream Function (Sv)

(new) **Supplementary Figure 2: Ventilation and overturning developing in the North Pacific is seen in various Earth System Models in response to North Atlantic freshwater forcing.** Meridional overturning stream function (Sv) in intermediate-complexity Earth System Models run under glacial-like boundary conditions either with (e-h) or without (a-d) freshwater perturbations to induce overturning in the North Pacific. For a quantification of the maximum overturning stream function in each case, see Supplementary Table 1. For the details of the models and their forcing, see Supplementary Table 2.

Maximum Meridional Overturning (Sv), North of 31 °N				
	UVic	c-GENIE	LOVECLIM	LOVECLIM-LGM
N. Pacific, Control (“UVic_ctrl”)	3.2	0.8	2.8	2.8
N. Pacific, Ventilated (“UVic_NP”)	14.8	14.4	14.4	3.7
N. Pacific Anomaly (Ventilated – Ctrl)	+11.6	+13	+11.6	+0.9
N. Atlantic, Control	14.2	14.2	20.8	21.8
N. Atlantic, Ventilated	1.3	0.1	5.4	12.9
N. Atlantic Anomaly (Ventilated – Control)	-12.9	-14.1	-15.4	-8.9

Supplementary Table 1. Model comparison of North Pacific and North Atlantic overturning strengths.

Maximum of the meridional overturning stream function north of 31 °N across the four models depicted in Figure 3. The cut off of 31 °N removes the influence of subtropical gyre circulation across all four models, but the results are otherwise insensitive to choice of latitude. Notably, despite the wide range in magnitudes of overturning anomalies, all models see a reduction of Pacific mid-depth carbon and nutrients in response to North Pacific ventilation (Fig. 3, Supplementary Fig. 3). Note also that in the ventilated LOVECLIM-LGM simulation, Pacific overturning expands in extent more so than magnitude over the control (Supplementary Fig. 2).

4. Methods and model description

R2 Comment 5/18: It is a bit confusing to find results of three different models (Fig.3) but only the model description of one model in 'Methods'. And please explain why different models are needed for this study and what the major differences between them (see above).

We are grateful to the reviewer for pointing out this confusion. In response, we have included a new Supplementary Table 2 (line ~148 in the Supporting Information, and copied below) that provides details on the crucial features of all three models (four simulations) including their original references for more in-depth assessment.

The motivation behind showing results from multiple models (Figure 3) was to demonstrate that a reduction of mid-depth carbon/nutrients is a robust response to ventilation, independent of model choice or set-up. That the same pattern emerges in a range of models spanning different configurations, resolutions, boundary conditions (though all are glacial-like), and ventilation strengths substantiates the link between ventilation and sub-surface carbon/nutrient content, which forms the basis of this study. We have added text that clarifies this purpose behind Figure 3 (lines ~150 and ~158-160) and also the rationale for carrying on with just UVic (per the edits referenced in response to the previous comment).

These edits are supplemented by additional text in Figure 3's caption (lines ~582-584 and ~590-593) summarizing the motivation behind this figure and the key differences between the models.

R2 Comment 6/18: It is not clear if the perturbation experiments with different models (e.g. Fig. 3) were all done in the exactly same way, since only UVic-ctrl and UVic-NP are described in the manuscript.

The edits above will contribute to addressing this issue. The perturbations experiments depicted in Figure 3 were all conducted similarly, applying either positive/negative freshwater forcing into the N. Atlantic/N. Pacific to induce changes in overturning in both basins. This is described in Figure 3's caption (lines ~586-590) and in the new Supplementary Table 2 (line ~148 in the Supporting Information).

R2 Comment 7/18: A brief description which components and processes of the marine carbon cycle are included in UVicESM would help to understand the analysis of model results. For example, changes in net primary production were discussed. But how it is calculated in the model? Is it only controlled by PO₄ as the only nutrient or are other nutrients involved as well? Which forms of carbon are considered in the model? I am not asking for a detailed description of the whole ESM but only for components and processes which are important for this study.

This information has now been added to the description of UVic in the Methods section (lines ~485-492). In brief, the UVic simulations use the marine ecosystem/biogeochemical model of Schmittner et al. (2008), in which net primary production is co-limited by nitrate, phosphate, and irradiance. Marine carbon chemistry follows Ocean Carbon-Cycle Model Intercomparison Project (OCMIP) (Orr et al., 1999) protocols. A full inorganic carbon cycle is simulated as well as organic carbon components, with particulate carbon being remineralized into both DIC and DOM pools.

Schmittner, A., Oschlies, A., Matthews, H. D., & Galbraith, E. D. (2008). Future changes in climate, ocean circulation, ecosystems, and biogeochemical cycling simulated for a business-as-usual CO₂ emission scenario until year 4000 AD. *Global biogeochemical cycles*, 22(1).

Orr, J., Najjar, R., Sabine, C. L., & Joos, F. (1999). Abiotic-howto. Internal OCMIP Report, LSCE/CEA Saclay, Gif-sur-Yvette, France, 1999.

R2 Comment 8/18: 'LGM-like boundary conditions' for the experiments need to be specified. And, is the atmospheric CO₂ interactively coupled (also for radiative forcing)? L206 mentioned 'more LGM-like simulations' from Rae et al. (2020). Please explain more in detail the differences between these and the simulations conducted in this study and why the simulations in this study were not conducted under the 'more LGM-like conditions'.

LGM-like boundary conditions

- “LGM-” or “glacial-like boundary conditions” refers to the key factors of glacial topography and albedo, orbital parameters, and atmospheric CO₂ levels in all cases for the models depicted in Figure 3. This information is now included in the new Supplementary Table 2 (line ~148 in Supporting Information) provided in response to comment #5 above. A fuller description of these boundary conditions for the UVic simulations specifically is now included in the Methods (lines ~500-503).

Is atmospheric CO₂ interactively coupled, also for radiative forcing?

- Yes, atmospheric CO₂ is fully coupled and radiatively active. This information has been included in the description of UVic in the Methods (lines ~487-488).

Explain differences between “LGM-like” simulations and those (UVic) in this study, and why didn’t run under more LGM-like conditions:

- We have improved the clarity on this point. The “more LGM-like” qualification the reviewer refers to (at line 206 in the previous version of the manuscript) refers exclusively to the magnitude of overturning (both Atlantic and Pacific) seen in the simulations. The simulations depicted in Figure 3 (and others from Rae et al. 2020 which we reference) are all run under glacial-like or LGM-like conditions and are thus comparable. The defining differences between them would be the strength (reduction) of overturning induced in the Pacific (Atlantic) (summarized in new Supplementary Table 2).
- We have revised the text to clarify this (line ~241), and we believe our choice of analyzing the more strongly-ventilated UVic simulation is supported by edits made to the text in response to comments #4 and #5 above.

R2 Comment 9/18: Where and how large is the area receiving freshwater hosing?

The freshwater hosing in UVic-NP was applied over a region of the N. Atlantic spanning 50-65°N and 55-10°W. This information is now provided in Supplementary Table 2 (line ~148 in the Supporting Information) and at relevant points in the main text (lines ~220-221) and Methods (lines ~498-499).

R2 Comment 10/18: Are model results of the last model year (without inter-annual variability) of both runs compared or the averages over some of the last years?

Averages over the last 10 years of simulation are compared between UVic-ctrl and UVic-NP. This information is provided in the Methods (lines ~503-504) and in the captions of Figures 3, 4, and 5 (lines ~599-600; ~613; and ~625, respectively), and Supplementary Figures 2-9 (in Supporting Information lines ~44-46, ~54-56; ~64-65; ~70-71; ~86; ~105-106; ~115; and ~126 respectively).

R2 Comment 11/18: Please clarify if the calculation of PCO₂ was done in the same way for all models.

Yes, PCO₂ was calculated in the same way for all models. This has been clarified in Figure 3’s caption (lines ~585-586).

	UVic	c-GENIE	LOVECLIM	LOVECLIM-LGM
Reference	Menviel et al., 2014	Rae et al., 2020	Same as UVic	Menviel et al., 2017
Resolution (latitude x longitude, depth layers)	1.8° x 3.6°, 20 depth layers	5° x 10°, 16 depth levels	3° x 3°, 20 depth layers	3° x 3°, 20 depth layers
Ocean Component	MOM v2	c-GENIE (a frictional geostrophic 3D ocean model)	CLIO	CLIO
Boundary Conditions	LGM-like (Last Glacial Maximum, ~21 kaBP) for glacial topography and albedo, orbital parameters, and	Glacial-like for radiative forcing consistent with major greenhouse gas concentrations (CO ₂ ,	LGM-like (Last Glacial Maximum, ~21 kaBP) for glacial topography, planetary albedo, orbital parameters,	LGM-like (first equilibrated under 35 kaBP boundary conditions and run transiently to 20

	atmospheric CO ₂ (220 ppmv)	CH ₄ , N ₂ O), planetary albedo, increased average ocean salinity, and atmospheric CO ₂ (278 ppm)	and atmospheric CO ₂ (191.85 ppmv)	kaBP) glacial topography and albedo, orbital parameters, and atmospheric CO ₂ (190 ppmv)
Forcing Style	N. Atlantic freshwater hosing	Reduced prescribed atmospheric freshwater flux from N. Atlantic to N. Pacific	N. Atlantic freshwater hosing	N. Atlantic freshwater hosing
Forcing Details	0.1 Sv freshwater for 1000 years into N. Atlantic (55-10°W, 50-65°N)	-0.28 Sv of freshwater forcing into N. Pacific over 5000 years	Same as UVic	0.05 Sv of freshwater forcing into N. Atlantic over 4000 years
Ventilation Strength (maximum overturning in N. Pacific, Sv)	14.8	14.4	14.4	3.7

Supplementary Table 2. The output depicted in Figure 3 comes from glacial-like simulations of various intermediate-complexity Earth System Models spanning varying configurations, resolutions, forcings, etc. All models depicted in Figure 3 were run under glacial-like boundary conditions and induced North Pacific ventilation and enhanced North Pacific Intermediate Water (NPIW) formation by manipulating freshwater forcing. In c-GENIE, the prescribed transfer of atmospheric freshwater from the Atlantic to the Pacific was reduced. In all other models, Atlantic overturning was suppressed by adding meltwater into the North Atlantic. Resulting oceanic and atmospheric teleconnections then led to enhanced NPIW formation. Beyond this, the models differ slightly in their details, which are summarized here. References: Menviel et al. (2014), Menviel et al. (2017), and Rae et al. (2020).

Meissner, K. J., Schmittner, A., Weaver, A. J. & Adkins, J. F. Ventilation of the North Atlantic Ocean during the Last Glacial Maximum: A comparison between simulated and observed radiocarbon ages. *Paleoceanography* 18, (2003).

Menviel, L., Timmermann, A., Mouchet, A. & Timm, O. Meridional reorganizations of marine and terrestrial productivity during Heinrich events. *Paleoceanography* 23, (2008).

Menviel, L., England, M. H., Meissner, K. J., Mouchet, A. & Yu, J. Atlantic-Pacific seesaw and its role in outgassing CO₂ during Heinrich events. *Paleoceanography* 29, 58–70 (2014).

Menviel, L. et al. Poorly ventilated deep ocean at the Last Glacial Maximum inferred from carbon isotopes: A data-model comparison study. *Paleoceanography* 32, 2–17 (2017).

Rae, J. W. B. et al. Overturning circulation, nutrient limitation, and warming in the Glacial North Pacific. *Sci Adv* 6, eabd1654 (2020).

Minor comments
R2 Comment 12/18: L152: Should be 'LOVECLIM?' This typo has been corrected (line ~153). R2 Comment 13/18: L246: If I understood correctly, UVic-ctrl was run under LGM conditions. Why is it compared with PI PCO₂? And, does Fig.1 show PI or present-day values? We acknowledge this was unclear. The text the reviewer references was drawing on the fact that the extent to which ventilation can reduce the carbon content of mid-depth Pacific waters is capped by the initial carbon concentration of the waters in question. In our control simulation, run under glacial-like conditions, the carbon content of North Pacific waters is lower than it would be under interglacial conditions. Thus, we suggest that ventilation could lead to even greater PCO₂ anomalies than presented here, if applied to a system transitioning out of interglacial conditions and into a glacial period. This is what we were trying to convey in these lines, and we have adjusted them accordingly (edits paragraph commencing at line ~284). Figure 1 depicts modern-day phosphate and nitrate (panels b, c). However, the PCO₂ depicted in panel a was calculated from modern DIC with anthropogenic carbon having first been removed, thus representing pre-industrial conditions. Nitrate and phosphate are not expected to have varied significantly between the modern and pre-industrial. Figure 1's caption (lines ~553-555 and ~560) has been adjusted to clarify that, while based on modern data, it is representative of a

pre-industrial/interglacial state. This is also clarified in the first reference made to this figure (line ~69-72).

R2 Comment 14/18: L276: 'varies by only...': increase or decrease? +1.5%?

Yes, this is an increase; this has been specified in the text (line ~317).

R2 Comment 15/18: L281: It would be helpful to see where biological productivity changes in a figure and how it compares to changes in PO₄.

A new Supplementary Figure 8 has been generated to address this. Phosphate is reduced across wide swaths of the Indo-Pacific as a result of North Pacific ventilation. Where NPP changes in the Indo-Pacific, it is reduced, consistent with nutrient signal. (NPP is virtually unchanged in regions of low biological productivity such as the sub-tropical gyres and Southern Ocean.) The text has been updated to reference this figure when discussing NPP changes (line ~326).

(New) Supplementary Figure 8. Negative anomalies in surface phosphate (PO₄³⁻) concentration pervade the Indian-Pacific basin and Southern Ocean, with net primary production (NPP) following suit. Surface phosphate (PO₄³⁻) concentration (mol kg⁻¹) (a, b) and net primary production rates (NPP, mol N m⁻³ yr⁻¹) (d, e) in UVic-ctrl (a, d), UVic-NP (b, e), and as anomalies (UVic-NP – UVic-ctrl) (c, f). Anomalies compare averages over the last 10 years of each simulation. Negative phosphate anomalies across the Indian-Pacific basin in response to North Pacific ventilation (c) drive corresponding reductions in NPP (f) (except in regions of already low biological production, such as subtropical gyres and the Southern Ocean, which see little or no anomalies in NPP).

R2 Comment 16/18: L353-355: Here it is not clear in which vertical layer the enhanced (or not) stratification and ventilation is meant. Enhanced stratification between intermediate waters and deep waters does not necessarily contradict enhanced ventilation of subsurface waters.

We have revised the text to more clearly specify that we are referring to upper-ocean stratification, which is what modulates the delivery of sub-surface nutrients to the surface. The depths relevant to this (in both our simulations and reality) are limited to the upper ~500 m. We have made edits to the text to reflect this more clearly (line ~425-426).

R2 Comment 17/18: L360: That models do not capture a feature (e.g. enhanced stratification) is not a strong argument that it unlikely happened. Furthermore, would it be better to involve more

recent results from PMIP4? Are model results of the last model year (without inter-annual variability) of both runs compared or the averages over some of the last years?

We agree with the reviewer that whether or not models capture a certain feature is not a strong indication of whether it did or did not happen. We have revised our language here (lines ~429-436) to reflect this, pointing out that there is a diverse range of model responses and that this thus underlines the importance of finding complementary mechanisms to reduce the Southern Ocean carbon/nutrient supply. We have also broadened the referencing to emphasize that reduced stratification is found in a variety of different models and is not exclusively a feature of PMIP2 and 3. We did also search to see if upper ocean stratification in the Southern Ocean had been systematically assessed in PMIP4, but at present we could not find such a study.

On the reviewer's final point, averages over the last 10 years of simulation are compared between UVic-ctrl and UVic-NP. References to edits made on this point can be found in our response to comment #10 above.

Reviewer #2 (Remarks on code availability):

R2 Comment 18/18: The model code is provided but the scripts to analyse model results. Since the model results will be provided by the corresponding author upon reasonable request, I was not able to run the model or test the scripts. If the model results will be provided in the revised manuscript, I will review the scripts for analysis.

The model output which we analyze is available to the reviewers on the Code Ocean capsule, alongside the scripts which perform the analysis. The model output will be made available for readers to access on a publicly accessible Zenodo repository (DOI 10.5281/zenodo.15276314). The output has already been deposited here and will be made publicly available upon publication.

The code performing our analysis is publicly available at https://github.com/Maddie-Sh/ShankleEtAl2025_NPacSObgc.

REVIEWER COMMENTS AND COAUTHORS' RESPONSES FOR:

“Southern Ocean CO₂ outgassing and nutrient load reduced by a well-ventilated glacial North Pacific”

Madison G. Shankle; Graeme A. MacGilchrist; William R. Gray; Casimir de Lavergne; Laurie C. Menviel; Andrea Burke; James W. B. Rae

Revisions Round 2 – 2025 June 20

REVIEWER #1

General Evaluation:

R1 Comment 1/6: The authors have provided thoughtful and detailed responses to each question. The revised figure captions are clearer and more informative. The concept of potential CO₂ is particularly interesting, as it offers a novel perspective on the capacity of different water masses to absorb atmospheric CO₂.

One important point concerns the interpretation of the timing of the numerical experiments in relation to paleoceanographic records. The authors analyze sediment data and highlight biogeochemical changes in the North Pacific and Southern Ocean during the Last Glacial Maximum (LGM). Meanwhile, the freshwater perturbation experiments in this study appear to be motivated by the intent to investigate the potential strengthening of NPIW during the LGM, as mentioned in the paragraph around Line 247.

If this is the case, does the study imply that many existing LGM control simulations fail to capture both the weakening of the AMOC and the strengthening of NPIW? This could potentially be an important message of the paper and might offer valuable insight into the current limitations of LGM modeling efforts.

While the manuscript is generally in good shape, a few points—particularly regarding clarity of model descriptions, consistency of units, and interpretation of key results—still require revision before the manuscript can be considered ready for publication.

We appreciate the positive feedback on our previous revisions as well as the new comments made here and below. Here the reviewer makes a particularly insightful observation of an important broader implication of our study. While we have not conducted a detailed review of existing LGM-like simulations and their treatment of compensating AMOC and NPIW changes, our results clearly underscore the need for such considerations in future work.

While historically not a primary focus of LGM modelling efforts, interest in glacial Pacific ventilation is growing (e.g., Rae et al., 2020; Millet et al. 2024) and continues to be an active area of debate (e.g., competing with the surface isolation hypothesis in explaining the “Polar Twins” proxy pattern, see references to Sigman in main text). Our study, having demonstrated how Pacific ventilation can both explain the proxy records and contribute to Southern Ocean outgassing changes, contributes to this debate a clear motivation for further targeted modelling efforts in this area. As the reviewer points out, accounting for the Pacific ventilation could provide valuable direction to guide future modeling efforts.

To reflect this key message, we have added text to the concluding paragraphs of the main text (lines ~499-503). We thank the reviewer again for this comment and the points raised below, which we feel have improve the strength and the clarity of our manuscript.

Rae, J. W., Gray, W. R., Wills, R. C. J., Eisenman, I., Fitzhugh, B., Fotheringham, M., ... & Burke, A. (2020). Overturning circulation, nutrient limitation, and warming in the Glacial North Pacific. *Science Advances*, 6(50), eabd1654.

Millet, B., Gray, W. R., de Lavergne, C., & Roche, D. M. (2024). Oxygen isotope constraints on the ventilation of the modern and glacial Pacific. *Climate Dynamics*, 62(1), 649-664.

Specific Comments:

R1 Comment 2/6: Line 184: “the long-term equilibrated response to ventilation of the North Pacific. Are the spin-up durations consistent across all models? For example, Supplementary Figure 4 shows that cGENIE was spun up for ~1000 years — clarification would be helpful.

- Yes, as the simulations presented in Figure 3 have come from different studies of differing motivations and study designs, they have not all been run to the same duration. Supplementary Table 2 (describing the models) has been updated to provide this information (line ~164 in the Supporting Information).
- (Note, in the process of addressing this comment, an inconsistency in model processing was discovered which caused the LOVECLIM output in **Figure 3c** and **Supplementary Figure 3c** to be displayed at the incorrect timestep. This has been updated to show the correct timestep (i.e., after 1000 years of 0.1 Sv freshwater forcing, same as UVic, and as described in the main text and Supplementary Table 1). This update has not perceptibly changed the PCO₂ and PO₄³⁻ anomalies depicted in these figures over the previous iteration, but it has shoaled the isopycnals depicted in them slightly. This correction has also slightly updated the maximum overturning stream function values displayed in **Supplementary Table 1** for the ventilated LOVECLIM case, with a slightly weaker Atlantic overturning and slightly stronger Pacific overturning (which is also reflected in an updated panel g of **Supplementary Figure 2**). Again, this update does not alter the scientific findings or conclusions of the manuscript.

R1 Comment 3/6: Line 227: Only LOVECLIM v1.1 is explicitly mentioned here, but the study also analyzes UVic ESCM v2.9. The overturning value of ~14.8 Sv corresponds to the UVic simulation, not LOVECLIM. Please clarify.

- This paragraph introduces the UVic simulations (termed “UVic-ctrl” and “UVic-NP” in our manuscript) that underpin our analysis. However, because they are discussed in the context of the study they were originally published in – Menviel et al. 2014, which presented results from both UVic and LOVECLIM – it became ambiguous in parts of this paragraph which model we were referring to.
- While much of our description applies to both models, our focus here is solely on UVic. We have revised the text (lines ~241-256) to clarify this, providing a clearer lead-in to the analysis of the UVic simulations that follows.

R1 Comment 4/6: Line 302: “Our results therefore perhaps represent a conservative estimate of the effect of North Pacific ventilation on North Pacific and Southern Ocean biogeochemistry.” That said, the degree of NPIW strengthening varies across models — UVic appears to exhibit the strongest response. For example, Supplementary Figure 6 shows a ~3000-year difference in deep-minus-surface age, which raises the question of whether such changes are also observable in foraminiferal records.

- This sentence specifically refers to the low-carbon bias in UVic-ctrl in the North Pacific, where modeled subsurface waters exhibit lower PCO₂ (~350-450 μatm) than observational estimate for interglacial-like conditions (>1000 μatm, based on GLODAP; Fig. 1). As such, our analysis does represent a conservative estimate, as modeled PCO₂ anomalies will always be capped by the starting PCO₂ content of waters in the control state, which UVic-ctrl underestimates.
- The reviewer is correct, however, that our results are not conservative based on the ventilation strength of UVic, and we qualify this better with revisions to the text near this sentence (lines ~321-323).
- Concerning ventilation strength, it is true that UVic ventilation is strong, but we don't expect this to impinge upon our results (with mid-depth carbon and nutrient reductions resulting from a range of ventilation strengths, see discussion starting at line ~170 in the text). While proxies cannot provide an exact number on overturning, they do indicate better ventilation of the North Pacific, and given that these are the oldest waters in the

global ocean, very large age changes (essentially transitioning from being among the ocean's oldest waters to much younger) are plausible. We put forward this point with edits to the figure caption of Supplementary Figure 6 (lines ~101-104 in the Supporting Information).

R1 Comment 5/6: Supplementary Figure 5: Please clarify the mechanism behind the positive air-sea CO₂ flux anomaly in the North Atlantic (UVic-NP minus UVic-ctrl). What physical or biogeochemical processes are responsible?

- In brief, the changes in air-sea CO₂ flux in the North Atlantic are due to the AMOC weakening and are therefore not linked to Pacific ventilation (nor representative of the LGM state) and thus lie beyond the scope of this paper. While emphasizing this point, we acknowledge the need to provide more detail describing this mechanism where relevant.
- Firstly, the flux anomalies in the North Atlantic are partly driven by decreased surface alkalinity due to dilution by the freshwater forcing applied here. Stronger stratification resulting from the freshwater input may also contribute to surface alkalinity decrease.
- Stronger stratification and a reduced AMOC also results in poorer ventilation of North Atlantic sub-surface waters. They consequently accumulate remineralized carbon – much like the modern North Pacific – forming a greater supply of sub-surface carbon to fuel outgassing.
- Finally, we also expect the reduced atmospheric pCO₂ (-6 ppm) between the two simulations to drive some of the anomalous outgassing signal, especially where changes in surface PCO₂ are minimal (compare regions of low-to-no surface PCO₂ anomaly in Supp. Fig. 5c to the more pervasive flux anomalies in Supp. Fig. 5f)
- We have added this discussion to the figure caption of Supplementary Figure 5 (lines ~75-85 in the Supporting Information) and also provide an overview to the reader where it comes up in the main text (lines ~432-434), referring them to the supplementary figures for more information.

R1 Comment 6/6: Supplementary Figure 8: Caption: "Phosphate, (mol kg⁻¹)" → Should be "μmol kg⁻¹"? "NPP, (mol N m⁻³ yr⁻¹)" → Should this be "mol N m⁻² yr⁻¹"? Please verify the units.

- Yes, "μmol kg⁻¹" is the correct unit for phosphate concentration; this typo in the figure caption has been fixed (line ~126 in the Supporting Information).
- For NPP, "mol N m⁻³ yr⁻¹" is the correct unit in this case. The NPP diagnostic defined within the model describes the change in molar N concentration in a grid cell due to NPP. Additional text in the figure caption (lines ~130-131 in the Supporting Information) has been added to clarify this.

REVIEWER #2

I much appreciate that the authors addressed all my comments and provided a lot more analyses, figures and model descriptions. However, based on the new material three major issues have been raised during the review.

We appreciate the reviewer's continued engagement and thoughtful assessment of our revisions. The newly raised issues are well-taken and highlight important points that we now address more thoroughly in our manuscript.

R2 Comment 1/3: 1) The central finding of this study is that a stronger NP ventilation during glacials lowers the carbon and nutrient supply to the surface Southern Ocean through NPIW, and thus contributes to glacial CO₂ drawdown. So far this study shows lower concentrations of carbon and nutrients in NPIW in simulations with well-ventilated NP, and also in intermediate waters in the South Pacific, and lower concentrations of carbon and nutrients in the SO surface waters. However, the main source of carbon and nutrients to the SO surface waters is the upwelling of Circumpolar Deep Water (CDW). NPIW flows southward and reaches the tropical surface by upwelling. It is not clear to me how NPIW can reach the SO surface directly. I believe that a stronger PMOC and thus transport of carbon and nutrient-poor surface waters to the deep ocean could contribute to a lower carbon and nutrient load in CDW and then affect the outgassing in the SO. But this should happen rather through the mixing of NPIW with deeper water masses and the deep circulation pathway and upwelling in the SO. Thus, the linkage between reduced carbon and nutrients in NPIW and SO surface waters need to be thoroughly discussed and more analysis of deep mixing and circulation which in my opinion might transport the signal in carbon and nutrients from NPIW to SO surface is necessary.

Additionally, the upwelling of CDW brings signals from other ocean basins as well, for example, the signal from the Atlantic which is strongly affected by AMOC weakening in this study. How large the contribution of NP ventilation is, needs to be better quantified.

If in simulations the SO surface water is supplied with carbon and nutrients in a different pathway than through CDW, this needs to be shown, explained and confirmed by proxy data.

- The reviewer has pointed out two inconsistencies in our terminology that would likely lead to confusion for readers.
- **(1) On our use of "NPIW":** The reviewer is correct that NPIW in the present day constitutes a relatively small overturning cell only reaching the tropics. However, this overturning is greatly expanded in strength, depth, and extent in our simulations (Supp. Fig. 2), helping to bring carbon and nutrient anomalies southwards. While the overturning does part of the work of bringing NPIW waters to the Southern Ocean, the more conclusive evidence of their connection are the mid-depth isopycnals that extend from the North Pacific to the high southern latitudes and outcrop in the Southern Ocean surface (as seen in Figs. 3 and 5, respectively). Anomalies of carbon and nutrients mainly reach the Southern Ocean diffusively along these isopycnals, consistent with recent advancements in the literature that highlight diffusive transport as a key mechanism in Pacific circulation (Holzer et al., 2021).
Holzer, M., DeVries, T., & de Lavergne, C. (2021). Diffusion controls the ventilation of a Pacific Shadow Zone above abyssal overturning. *Nature Communications*, 12(1), 4348.
- However, given its major change in character, we acknowledge that it is confusing to continue to refer to this expanded overturning in our ventilated simulation as "NPIW". **To avoid ambiguity, we now use "GNPIW" ("glacial NPIW") to refer to the expanded North Pacific overturning cell and mid-depth waters in our ventilated simulation (lines ~212, ~253, ~287, ~400).**
- **(2) CDW and altered deep-ocean carbon supply:** The reviewer also points out correctly that it is CDW that upwells into the Southern Ocean, not NPIW.
- CDW itself forms through mixing of PDW, IDW (Indian Deep Water), and some Southern Ocean-sourced water that is not dense enough to become AABW. This mixture forms Upper Circumpolar Deep Water (UCDW) – the upwelling water and primary source of carbon and nutrients to the Southern Ocean that is the focus of our

study – while Lower Circumpolar Deep Water forms from the mixing of NADW with the same Southern Ocean-sourced water. While CDW thus indeed forms from some underlying waters below in this way, the underlying waters are not the chief source of carbon to CDW.

Talley, L. D. (2013). Closure of the global overturning circulation through the Indian, Pacific, and Southern Oceans: Schematics and transports. *Oceanography*, 26(1), 80-97.

- Rather, the high carbon content of PDW (and ultimately CDW) primarily results from its poor ventilation of and strong isolation from the surface. Without any exchange with carbon-poor surface waters, this allows significant amounts of remineralized carbon to accumulate in this region over time. The underlying water, by contrast, is much less important in the carbon supply to these waters, and altering PDW/CDW carbon content by altering the carbon content of the water below it would require a substantial reduction in deep ocean carbon, which is not a feature of glacial proxy records.
- Instead, in our study we find that the depths of normally carbon-rich PDW gets replaced by expanded (low-carbon) GNPIW in response to North Pacific ventilation, ultimately reducing the amount of carbon feeding into the Southern Ocean.
- We believe the edits made in response to point 1 above will contribute to highlighting the importance of these mid-depths waters and their carbon content as the main leverage over carbon/nutrient supply to the Southern Ocean, as also recently highlighted by Chen et al. (2022). **We have added additional text, however, discussing this new circulation regime in more detail at relevant points in the text (lines ~73-78, ~165-167, ~211-216).**

Chen, H., Haumann, F. A., Talley, L. D., Johnson, K. S., & Sarmiento, J. L. (2022). The deep ocean's carbon exhaust. *Global Biogeochemical Cycles*, 36(7), e2021GB007156.

- “The upwelling of CDW brings signals from other ocean basins as well, for example, the signal from the Atlantic which is strongly affected by AMOC weakening in this study. How large the contribution of NP ventilation is, needs to be better quantified.”:
- In UVic-NP, the Atlantic does supply more carbon-rich waters to the Southern Ocean due to its weakened AMOC, poorer ventilation, and build up of remineralized carbon (though note the strongest PCO_2 anomalies do not effectively upwell into the Southern Ocean; see figure below). That the Southern Ocean surface still exhibits reduced carbon and nutrients despite this Atlantic signal, however, speaks to the dominant role played by Pacific biogeochemistry in shaping this result.
- While positive PCO_2 anomalies from the Atlantic are of comparable magnitude to negative anomalies from the Pacific, the latter delivers a larger volume of water to the Southern Ocean, again consistent with its dominant role over Southern Ocean biogeochemistry. Beyond this, an exact quantification of Atlantic versus Pacific contributions to the Southern Ocean carbon/nutrient budget is not warranted (as these simulations were not designed to be interpreted as exact reproductions of the LGM ocean state), but the above considerations provide a robust qualitative demonstration of our conclusions. Text stating as much and addressing the point of Atlantic carbon and overturning has been added at the relevant discussion (lines ~86-91 in the Supporting Information).

Figure for response document: PCO_2 anomaly up the Atlantic basin (along $30^\circ W$), UVic-NP – UVic-ctrl.

R2 Comment 2/3: 2) In SI, Figure 2 and Table 1 show that the perturbed simulation with UVic is quite far from a LGM state and rather represents the onset of deglaciation, such as Heinrich Stadial 1. I am wondering how the results from this simulation can help to understand mechanisms driving the glacial CO₂ drawdown. The authors addressed this in the revised manuscript that they like to focus on the impact of ventilated NP on the SO biogeochemistry. However, the ocean is connected which is particularly true for the Southern Ocean where waters from all ocean basins are joined and mixed, as the authors also pointed out in the introduction. If the focus of the study is the contribution of NP ventilation to glacial CO₂ drawdown, a simulation like LOVECLIM-LGM with a stronger PMOC and a weaker AMOC but not an AMOC shutdown would be more plausible. The anomaly of carbon and nutrients in the SO surface/upwelling waters (Fig.3 and SI Fig.2) however, seems to be negligibly small, not supporting the hypothesis that a well-ventilated GLACIAL NP could account for the reduction of carbon and nutrients in the SO surface.

We thank the reviewer for their thoughtful probing of our simulations. We fully acknowledge that the perturbed UVic simulation in one aspect (its Atlantic overturning) departs from what may be considered an “LGM-like” state, but it remains a robust representation of a glacial-like state in other key aspects. More importantly, however, it was not our intention to replicate the LGM *per se*, but rather to explore the mechanistic role of North Pacific ventilation in influencing basin-wide and Southern Ocean biogeochemistry. In this sense, the simulations we present – spanning various configurations and ventilation strengths – are ideally suited to our purpose. That the same broad pattern emerges across such varied ventilation strengths is the very thing that enables us to interpret the results without over-interpreting the details of any single model too closely.

It is also important to state that, though the existence of a ventilated glacial North Pacific is increasingly supported by proxy data, the magnitude of North Pacific ventilation at the LGM is far from constrained. While the LOVECLIM-LGM simulation is presented as an estimate of a more glacial-like magnitude of overturning (provided as context for comparison against the more strongly-ventilated simulations in Figure 3), this simulation may in fact underestimate glacial Pacific overturning (~4 Sv, Supp. Table 1). For example, Rae et al. (2020) find that an overturning of ~8 Sv yields the best model-data fit in c-GENIE (that simulation being a counterpart to the more-strongly ventilated c-GENIE simulation presented in Fig. 3b, see comparison below) and furthermore ventilates the upper 2km of the water column of the North Pacific with GNPIW (Millet et al., 2024), much like the rest of the simulations presented in Figure 3. This illustrates a fundamental challenge in modelling past climates: that no single model is likely to definitively reproduce past dynamics, and it is best practice to interpret broad trends and patterns consistent across multiple models. Accordingly, our goal was not to propose a precise quantitative scenario (in UVic or any of the other simulations), but rather to explore a plausible mechanism that emerges as a robust feature across configurations and ventilation strengths. Thus, the LOVECLIM-LGM results should be viewed as indicative rather than definitive. Furthermore, even if its magnitude were overestimated, this does not rule out North Pacific ventilation as a meaningful mechanism operating under past glacial states and indeed the simulations of Rae et al. (2020) show striking reductions in subsurface nutrients and carbon in North Pacific waters under even modestly enhanced overturning.

There are other factors besides overturning strength to be considered as well. Most notable is the fact that the UVic control simulation, run under glacial-like conditions, underestimates baseline carbon storage (pre-ventilation) in the subsurface North Pacific (see discussion in main text at lines ~315-321). This caps the PCO₂ anomalies achievable through ventilation. A similar low-carbon bias exists in the LOVECLIM and LOVECLIM-LGM simulations. Correcting this bias would likely amplify the anomalies, meaning the relatively weak overturning in LOVECLIM-LGM may very well be capable of producing stronger, more coherent signals.

Finally, outside of modeling studies, proxy evidence continues to support a role for North Pacific ventilation in the glacial ocean (Rae et al., 2020; Rafter et al., 2022; Millet et al., 2024). Our results provide a neat explanation of the shared Polar Twins proxy pattern between the

North Pacific and Southern Ocean (Haug & Sigman, 2009), indirectly supporting the existence of North Pacific ventilation – an important result given that a satisfactory explanation for this pattern remains elusive. Emerging data challenge the traditional view of suppressed high-latitude ventilation: for example, stronger Ekman suction and a saltier subpolar gyre at the LGM (Gray et al., 2020) argue against the long-standing surface stratification hypothesis for the North Pacific. Such inconsistencies between emerging proxy evidence and earlier concepts of a poorly-ventilated glacial North Pacific further motivate our exploration of alternative processes such as the one we propose.

In summary, while anomaly magnitude and structure vary across the presented simulations and none may perfectly replicate an LGM-like state, this was not our goal. Instead, this suite of models serves as a useful platform to investigate the physical and biogeochemical impacts of enhanced North Pacific ventilation. Given the inherent challenges of modeling past climates and the lack of consensus among even advanced models, we argue that our proposed mechanism is as plausible as others in the literature and warrants further investigation. As such, our findings offer mechanistic insights that we assert remain meaningful despite model limitations.

Even so, the discussion made here contains valuable detail and context that should have been included in the main text, to better support our interpretation and discussion of the results. For this reason we much appreciate the reviewer's engagement. We are now more transparent on this point and have provided additional discussion at relevant points in the manuscript, which may be found at lines ~174-182 in the main text and in the captions of Figure 3 (lines ~623-625) and Supplementary Table 1 (lines ~158-163 in the Supporting Information).

Rae, J. W., Gray, W. R., Wills, R. C. J., Eisenman, I., Fitzhugh, B., Fotheringham, M., ... & Burke, A. (2020). Overturning circulation, nutrient limitation, and warming in the Glacial North Pacific. *Science Advances*, 6(50), eabd1654.

Rafter, P. A., Gray, W. R., Hines, S. K., Burke, A., Costa, K. M., Gottschalk, J., ... & DeVries, T. (2022). Global reorganization of deep-sea circulation and carbon storage after the last ice age. *Science Advances*, 8(46), eabq5434.

Millet, B., Gray, W. R., de Lavergne, C., & Roche, D. M. (2024). Oxygen isotope constraints on the ventilation of the modern and glacial Pacific. *Climate Dynamics*, 62(1), 649-664.

Haug, G. H., & Sigman, D. M. (2009). Polar twins. *Nature Geoscience*, 2(2), 91-92.

Figure for response document: PCO₂ anomaly (along 160°W, as in Fig. 3) in response to “LGM-like” ventilation of the North Pacific (~8 Sv) in c-GENIE (Rae et al., 2020), for comparison with the more strongly-ventilated c-GENIE simulation from that study (our Fig. 3b) and another estimate of LGM-like ventilation from LOVECLIM-LGM (~4 Sv, our Fig. 3d) below.

R2 Comment 3/3: 3) Figure 9 in SI show the air-sea CO₂ fluxes. I guess that the correct unit is molC/yr as in the caption, not as at the Y-axis.

Yes, molC/yr is the correct unit; this has been fixed on the y-axis of Supp. Fig. 9 (line ~137 in Supporting Information).

The global flux (the sum of the area below or above the curve) seems to be a net outgassing. It is even clearly the case in the control simulation. Is it still in a transient state? This is also reflected in SI Figure 5 where the absolute fluxes are shown. Although it is stated in Line 518-520: 'CO₂ was set to be prognostic in these simulations, though the LGM-like background state was first established by forcing the model under LGM CO₂ (192 ppm) until a quasi-equilibrium was reached'. Furthermore, a CO₂ decrease of 6 microatm is reported in the study from the control to perturbation experiment. How can it be explained by the strong outgassing trend from the ocean? If the air-sea and air-land gas exchange are both considered and coupled in the ESM simulations, could changes in land carbon explain this CO₂ decrease? For these questions, the temporal evolution of pCO₂ in the atmosphere during the entire simulation would help to check if the land-ocean-atmosphere gas exchange approaches an equilibrium, and the same for the total land carbon and ocean carbon pool.

In the process of addressing this comment, we discovered that SF9 was not plotting the correct model diagnostic for flux, which is what made the ocean appear to be net-outgassing as the reviewer observed. We have corrected this and SF9 now (correctly) shows that overall ocean fluxes are roughly in equilibrium with the atmosphere. Updating this diagnostic introduced minor changes to SF9 and SF5 but has not significantly affected the main signals, anomalies, or our discussion in the main text.

Nevertheless, the reviewer is right to ask about the major carbon reservoirs and below we provide the time-evolution of the land, ocean, and atmospheric carbon inventories. The system is very near equilibrium. Residual trends in these reservoirs are minimal, varying by <0.3ppm/100 years in the atmosphere and <1PgC/100 years in the atmosphere and ocean. We therefore consider the signals presented in the main text to be representative of a fully equilibrated state and for land-ocean-atmosphere gas exchange to be effectively balanced at the end of the simulation.

Most critically, we observe an accumulation of carbon in the ocean over time as North Pacific ventilation develops, and we can confirm that the 6 ppm reduction in atmospheric CO₂ is not driven by changes in the land carbon pool since this reservoir actually loses carbon over the course of the simulation.

We have revised the text (lines ~245-249) to state that the major carbon reservoirs have roughly stabilized and also briefly discuss the land carbon reservoir changes, supporting the

conclusion that the ocean drives atmospheric CO₂ drawdown (lines ~436-437 and in the caption of Supp. Fig. 9, lines ~144-145 in the Supporting Information).

Figure for response document: Time evolution of atmospheric CO₂ and the ocean and land carbon inventories over the course of the ventilated UVic-NP simulation.

REVIEWER COMMENTS AND COAUTHORS' RESPONSES FOR:

“Southern Ocean CO₂ outgassing and nutrient load reduced by a well-ventilated glacial North Pacific”

Madison G. Shankle; Graeme A. MacGilchrist; William R. Gray; Casimir de Lavergne; Laurie C. Menviel; Andrea Burke; James W. B. Rae

Revisions Round 3 – 2025 August 22

Reviewer #1 (Remarks to the Author):

The noteworthy result of this study is that it demonstrates how the strengthening of deep circulation in the North Pacific reduces the supply of nutrients and carbon to the Southern Ocean through remote effects, thereby suppressing CO₂ outgassing from the Southern Ocean. Importantly, the study offers a new perspective on past North Pacific circulation, supported by proxy evidence.

This work is significant to the field as it addresses a gap in our understanding of the role of North Pacific deep circulation in regulating Southern Ocean CO₂ outgassing.

The results and figures presented are clear and, following revision, effectively support the authors' conclusions and claims.

There are no flaws in the data analysis, interpretation, or conclusions that would prohibit publication. The methodology is sound, meets the expected standards of the field, and is described in sufficient detail to allow for reproducibility.

It is my understanding that clarifying the specific periods during which this mechanism operated in the past, as well as the timescales involved, remains an important subject for future investigation.

Reviewer #2 (Remarks to the Author):

The authors provided thorough and insightful responses to my comments, clarifying key terminology, correcting and supplemented figures with deeper analyses, and clearly outlined the goals and limitations of the study. I fully agree that the perturbation experiments presented here make a valuable contribution to understanding how stronger North Pacific ventilation influences carbon and nutrient supply to the surface Southern Ocean.

While I remain not fully convinced that these simulations directly help to explain glacial CO₂ drawdown—given that they may not fully represent a glacial ocean state, the study nevertheless offers important novel perspectives and useful directions for future research.

For these reasons, I support the publication in Nature Communications.

Minor comments:

L23: '...global nutrients, carbon cycling' to 'global nutrient and carbon cycling'

L35: 'glacial-interglacial CO₂ change' to 'glacial-interglacial CO₂ variability'

L170: '...expanded NPIW cell has expanded and ...': delete the second 'expanded and'?

L180: 'Rae et al. (2020) find' to 'Rae et al. (2020) found'

L254 and L258: are the parentheses before 'After the 1000-year' and after 'a fully equilibrated state' necessary?

We are highly appreciative of the reviewers' feedback and are pleased they find the present work suitable for publication. Their positive comments on the manuscript are appreciated, and we agree with their identification of points of research which future studies may address (such as the timescales on which our proposed mechanism may have operated, and its place among other features of the glacial ocean state). It is our hope that the present study both motivates and has laid the groundwork for such work.

We have amended the typographical errors flagged by Reviewer 2, and we thank the reviewers again for their excellent work and review of our study. We look forward to seeing this work published.